# Synthesis of the Carbohydrate Moiety of Glycoproteins from the Parasite *Echinococcus granulosus* and Their Antigenicity against Human Sera

**DOI:** 10.3390/molecules26185652

**Published:** 2021-09-17

**Authors:** Noriyasu Hada, Tokio Morita, Takashi Ueda, Kazuki Masuda, Hiromi Nakane, Mami Ogane, Kimiaki Yamano, Frank Schweizer, Fumiyuki Kiuchi

**Affiliations:** 1Faculty of Pharmaceutical Sciences, Tokyo University of Science, Chiba 278-8510, Japan; 2Faculty of Pharmacy, Keio University, Tokyo 105-8512, Japan; morita.tokio@keio.jp (T.M.); zacro.ueda@gmail.com (T.U.); masudakazuki0000@gmail.com (K.M.); beautifulocean.june@gmail.com (H.N.); mamium-07@hotmail.co.jp (M.O.); kiuchi-fm84@keio.jp (F.K.); 3Hokkaido Institute of Public Health, Sapporo 060-0819, Japan; yamano@iph.pref.hokkaido.jp; 4Department of Chemistry and Biochemistry, Faculty of Science, University of Manitoba, Winnipeg, MB R3T 2N2, Canada; Frank.Schweizer@umanitoba.ca

**Keywords:** glycoprotein, *Echinococcus granulosus*, host-parasite interaction, stereocontrolled synthesis, biotin probe

## Abstract

Stereocontrolled syntheses of biotin-labeled oligosaccharide portions containing the carbohydrate moiety of glycoprotein from *Echinococcus granulosus* have been accomplished. Trisaccharide Galβ1-3Galβ1-3GalNAcα1-R (**A**), tetrasaccharide Galα1-4Galβ1-3Galβ1-3GalNAcα1-R (**B**), and pentasaccharide Galα1-4Galβ1-3Galβ1-3Galβ1-3GalNAcα1-R (**C**), (R = biotinylated probe) were synthesized by stepwise condensation and/or block synthesis by the use of 5-(methoxycarbonyl)pentyl 2-azido-4,6-*O*-benzylidene-2-deoxy-α-d-galactopyranoside as a common glycosyl acceptor. The synthesis of the tetrasaccharide and the pentasaccharide was improved from the viewpoint of reducing the number of synthetic steps and increasing the total yield by changing from stepwise condensation to block synthesis. Moreover, hexasaccharide **E**, which contains the oligosaccharide sequence which occurs in *E. granulosus*, was synthesized from trisaccharide **D**. We examined the antigenicity of these five oligosaccharides by an enzyme-linked immunosorbent assay (ELISA). Although compounds of **C**–**E** did not exhibit antigenicity against cystic echinococcosis (CE) patient sera, compounds **B**, **D**, and **E** showed good serodiagnostic potential for alveolar echinococcosis (AE).

## 1. Introduction

In the course of our studies on natural oligosaccharides from invertebrate animal species, we are interested in the unique glycosphingolipids (GSLs) and glycoproteins (GPs) found in various Protostomia phyla and we have synthesized these oligosaccharides in order to elucidate biological functions [1,2,3,4,5,6,7,8,9,10,11,12,13,14,15,16,17,18]. In particular, our interests have been focused on the unique oligosaccharide structures of GSLs and GPs found in several parasites including *Echinococcus multilocularis* [2,7,8,11,12,13,15], *Schistosoma mansoni* [5,10], *Ascaris suum* [1,17], and *Toxocara canis* [6]. Among them, GSLs and GPs from *E. multilocularis* have attracted our special attention. *E. multilocularis* is a parasite, which belongs to the class Cestoda of the phylum Platyhelminthes and causes alveolar echinococcosis (AE), a severe parasitic zoonosis that can be fatal without appropriate treatment. In 1992, Persat et al. reported [19] that sera from AE patients recognized a neutral glycosphingolipid fraction from *E. multilocularis* and determined the structures of some of the glycosphingolipids isolated from this fraction. Hülsmeier et al. reported [20] in 2002 that Em2, an antigen extracted from *E. multilocularis*, is a mucin-type glycoprotein. Based on this information, we synthesized four glycosphingolipids including Galβ1→6Galβ1-Cer as a core structure [15] and five carbohydrate structures of glycoproteins including a reducing terminal Galβ1→3GalNAcα1—which is a core 1 of mucin-type *O*-glycans of *E. multilocularis* [8,11], and examined antigenicity of the pure compounds by ELISA for their serodiagnostic potential [7,12,13].

More recently, Díaz et al. reported [21] that the extracellular matrix of a related cestode *Echinococcus granulosus* contains a novel mucin-type *O*-glycan capping motif consisting of Gal*p*α1-4Gal linkages at the non-reducing end (Figure 1). *E. granulosus* is a parasitic cestode causing cystic echinococcosis (CE) in intermediate hosts like humans. The adult worm lives in the small intestine of a carnivore (definitive host), and the intermediate larval stage can infect a wide range of mammal species—including humans—that acquire the infection through accidental ingestion of eggs. Díaz et al. elucidated in full the major glycan of the *E. granulosus* laminated layer (LL) and these are conventional core 1 and 2 *O*-glycans modified with Galα1-4Gal that are linked to three kinds of carbohydrates (Gal, GalNAc, and GlcNAc). Based on this information, we report here on the synthesis of the biotinylated glycan portions **A**–**E** (Figure 1 and Figure 2) of the glycoprotein antigen of *E*. *granulosus* in order to elucidate the interactions between the oligosaccharides and sera, and the structure-activity relationships involved in the antigen recognition. Compound **D** is a synthetic intermediate derived from the process of synthesizing compound **E**.

## 2. Results and Discussion

### 2.1. Syntheses of the Target Oligosaccharides ***A***, ***B***, and ***C***

The synthetic strategy for oligosaccharides **A**–**C** is shown in Figure 3 (NMR spectra provided in the Appendix A).

Suitably protected monosaccharide derivatives (**4**, **5**, **9**, and **12**) were chosen as building blocks. The 5-(Methoxycarbonyl)pentyl group was chosen as the temporary protecting group for all the target compounds to ensure future conjugation with biotin for the ELISA assay as previously shown by us [8]. The synthetic routes for target compounds **A**–**C** are outlined in Figure 1, Figure 2, Figure 3, Figure 4, Figure 5 and Figure 6. Initially, disaccharide acceptor **8** was prepared in a way that serves as a common acceptor for the syntheses of compounds **A**, **B**, and **C**. Disaccharide **8** was prepared from thioglycoside donor **3**, which was obtained from phenyl-1-thio-β-d-galactopyranoside (**1**) by regioselective 2-naphthylbenzylation of the in situ prepared stannylidene derivative with 2-naphthylbenzyl bromide (NapBr) and tetrabutylammonium bromide followed by benzoylation (Figure 1). The glycosylation of **3** with **5** [8] in the presence of *N*-iodosuccinimide (NIS)/trifluoromethanesulfonic acid (TfOH) [22] and AW-300 molecular sieves (MS AW-300) in CH_2_Cl_2_ afforded desired disaccharide **6** in 71% yield. The nature of the new β-glycosidic linkage was determined by the vicinal coupling constant of the anomeric proton (H-1 of Gal, δ = 5.11 ppm, *J* = 7.9 Hz). Condensation of **5** with the 3-*O*-chloroacetyl (ClAc) donor **4**, which was prepared by selective removal of the 3′-*O*-NAP group in **3** with 2,3-dichloro-5,6-dicyano-*p*-benzoquinone (DDQ) followed by chloroacetylation afforded desired disaccharide (**7**) in higher yield (89%). Oxidative removal of the 3′-*O*-NAP group in **6** with DDQ gave disaccharide acceptor **8** in 63% yield and removal of the 3′-*O*-ClAc group in **7** with thiourea produced the same acceptor **8** in 84% yield. Comparing the protecting groups at the 3’-position, the ClAc group consistently gave higher yields than the NAP group in both, glycosylation and deprotection steps in the synthesis of **8**. The glycosylation of disaccharide acceptor **8** with thioglycosyl donor **9** in the presence of NIS/TfOH and MS AW-300 in CH_2_Cl_2_ afforded desired trisaccharide (**10**) in 78% yield. The nature of the new glycosidic linkage was determined as β by the vicinal coupling constant of the anomeric proton (H-1 of Gal, δ = 4.69 ppm, *J* = 8.1 Hz). Selective removal of the 4″-*O*-ClAc protecting group from **10** was achieved by thiourea to produce trisaccharide acceptor **11** in 85% yield, which was the direct precursor of compound **A** and used directly for the next glycosylation step of compound **B**. The Glycosylation of **11** with **12** [23] using NIS/TfOH and MS AW-300 in CH_2_Cl_2_ produced desired tetrasaccharide (**13**) in 85% yield. The newly formed α-glycosidic linkage (H-1 of Galc, δ = 4.86 ppm, *J* = 2.6 Hz) was confirmed by ^1^H NMR spectroscopy.

Global deprotection of the precursors for **A** and **B** was performed by a combination of protection/deprotection steps. Initially, we attempted the simultaneous reduction of the azido group and removal of the benzyl protecting groups in **13** by catalytic hydrogenolysis with 10% Pd/C. However, since this conversion was not successful, we studied stepwise conversion. Selective hydrogenolysis using 10% Pd/C of the azido group in the presence of ammonia followed by hydrogenolytic cleavage of the benzyl groups over 10% Pd/C in acetic acid and exposure to Ac_2_O in pyridine resulted in the *O*- and *N*-acylation. However, the benzylidene acetal was not removed under these conditions. Thus, the benzylidene acetal was removed with TsOH followed by *O*-acetylation with Ac_2_O in pyridine to afford **14** in 51% yield after five steps. After deacetylation of **14** under Zemplén conditions, 5-(methoxycarbonyl)pentyl glycoside **15** was converted into the ethylenediamine monoamide by exposure to ethylenediamine and conjugated to biotin using our previously established methodology [8] to afford tetrasaccharide-biotin conjugate **A** in 67% yield after column chromatographic purification on Sephadex LH-20 (Figure 2). 

Tetrasaccharide **B** was synthesized in a multi-step sequence as outlined in Figure 3. At first, selective removal of the DTBS group in **13** was achieved with HF/pyridine and the benzylidene acetal was removed by acidic hydrolysis followed by acetylation with acetic anhydride in pyridine to afford **16** in 65% yield. Then, the azido-group was converted to an acetamido moiety by reduction with PPh_3_ followed by *N*-acylation with Ac_2_O in pyridine and debenzylation by catalytic hydrogenolysis using Pearlman’s catalyst followed by *O*-acetylation to produce protected tetrasaccharide **17** in 65% yield. The deacetylation of **17** under Zemplén conditions yielded unprotected tetrasaccharide **18** in 60% yield. Compound **18** was then used for ligation to biotin to provide target tetrasaccharide **B** after column chromatographic purification on Sephadex LH-20 (Figure 3). 

Originally, it was planned to prepare pentasaccharide **C** by elongation from previously prepared disaccharide acceptor **8** and thioglycoside donor **4**. However, preparation of the intermediate trisaccharide **19** by glycosylation of thioglycoside donor **4** with disaccharide acceptor **8** by NIS/TfOH-promoted activation was not successful. Similarly, the glycosylation of donor **20** [5] as well as donor **22** [24] with disaccharide acceptor **8** was also not successful in the presence of NIS and TfOH. The latter reaction afforded undesired α-glycosylated trisaccharide **24** in 67% yield (Figure 4). These results suggest that **4** and **20** are unreactive (mismatched) donors to react with disaccharide acceptor **8**. On the other hand, donor **22** displays superior reactivity but exclusively produces undesired α-anomer because of the absence of a C-2 acyl neighboring group participation. 

The failure to generate trisaccharide intermediates **19**, **21**, and **23** forced us to modify our original synthetic strategy as outlined in Figure 4. Compound **20** was selected as a new glycosyl donor of the di- and trisaccharides and the elongation of the carbohydrate chain was repeated by glycosylation and dechloroacetylation as shown in Figure 5 and Figure 6. The glycosylation of the acceptor **5** with **20** in the presence of NIS/TfOH provided disaccharide **25** in 76% yield. The disaccharide acceptor **26** was obtained in 76% yield from **25** after treatment with thiourea, which was used directly for the next glycosylation step. Trisaccharide derivative **27** was obtained by glycosylation of glycosyl donor **20** with **26**. The deprotection of the ClAc group in **27** was performed as described for compound **26** to provide trisaccharide acceptor **28** (Figure 5). 

The coupling of acceptor **28** to thioglycoside donor **9** afforded protected tetrasaccharide **29** in 53% yield. Selective removal of the 4-*O*-ClAc group in **29** by thiourea afforded tetrasaccharide acceptor **30** in 97% yield. In order to prepare the Galα1-4Gal-sequence with high α-stereoselectivity, we selected 4,6-*O*-di-*tert*-butylsilylene (DTBS)-protected galactose donor **12**. Previous studies have indicated that DTBS-protected galactose donors induce high α-selectivity in glycosylation reactions [23]. NIS/TfOH-promoted activation of thioglycoside donor **12** and coupling to tetrasaccharide acceptor **30** generated protected pentasaccharide **31** in 68% yield (Figure 6). 

Although we were able to achieve the synthesis of the desired protected target pentasaccharide using the stepwise elongation approach, the amount obtained was not sufficient to undergo global deprotection. As a result, we investigated a block synthesis approach in which a non-reducing terminal disaccharide derivative was synthesized in advance and condensed with the reducing end terminal di- and tri-saccharide derivatives to synthesize the tetra- and the penta-saccharides (Figure 5). 

Disaccharide donor **36** was obtained by using the synthetic strategy outlined in Figure 7. At first, the glycosylation of the known monosaccharide acceptor **32** [25] with monosaccharide donor **33** (commercially available) afforded benzylgroup-protected disaccharide **34** in 79% yield. Removal of the benzyl protecting groups of **34** by catalytic hydrogenation over 10% Pd-C in THF-MeOH followed by *O*-acetylation produced protected disaccharide **35**. Selective removal of the 2-(trimethylsilyl)ethyl (TMS-ethyl) group in **35** with TFA, followed by exposure of resulted hemiacetal to CCl_3_CN and 1,8-diazabicyclo [5,4,0]-7-undecene (DBU) afforded corresponding α-trichloroacetimidate donor **36** in 84% yield (Figure 7). 

The glycosylation of acceptor **26** with donor **36** in the presence of TMSOTf and MS AW-300 in CH_2_Cl_2_ afforded desired tetrasaccharide **37** in 34% yield. The nature of the new β-glycosidic linkage was determined by the vicinal coupling constant of the anomeric proton (H-1 of Galb, δ = 5.01 ppm, *J* = 8.1 Hz). The deprotection and biotinylation of **37** proceeded in excellent yield to give target tetrasaccharide **B** (Figure 8). 

Pentasaccharide **C** was obtained in moderate yields from trisaccharide acceptor **28** and disaccharide donor **36** in a similar manner (Figure 9). Deprotection and biotinylation of pentasaccharide **40** were performed as described for compound **B** to provide target trisaccharide **C** in an excellent 85% yield (Figure 9).

### 2.2. Syntheses of the Target Oligosaccharides ***D*** and ***E***

We next devised a synthetic strategy for trisaccharide **D** and hexasaccharide **E** as shown in Figure 6 (NMR spectra provided in the Appendix A). Trisaccharide **D** constitutes the partial structure of hexasaccharide **E**. Trisaccharide **44** served as starting material for the preparation of **D**. Trisaccharide **45** was prepared by condensation of 2,6-dimethyl-thiophenyl-trisaccharide donor **44** [11] with 5-(methoxycarbonyl)pentyl alcohol in the presence of NIS/TfOH and MS AW-300 in CH_2_Cl_2_ in 89% yield. The reduction and *N*-acetylation of the Troc groups of **45** with Zn-Cu THF/AcOH/Ac_2_O followed by debenzylation with catalytic hydrogenolysis over 10% Pd/C in MeOH and acetylation afforded **46**. Deacylation of **46** followed by biotinylation of **47** was performed as described above to provide target trisaccharide **D** (Figure 10).

Compound **E** is a hexasaccharide which combines two trisaccharide components: Galα1-4Galβ1-4GlcNAc and Galβ1-3Galβ1-3GalNAc. The former component can be conveniently installed using synthetic intermediate **44** while the latter component **48** was obtained from the regioselective reductive ring-opening of the benzylidene acetal in compound **10** as a glycosyl acceptor (Figure 11). The glycosylation of **44** with **48** in the presence of NIS/TfOH and MS AW-300 in CH_2_Cl_2_ afforded desired disaccharide (**49**) in 94% yield. The new β-glycosidic linkage was confirmed by ^1^H NMR using the coupling constant of H-1 of GlcN (δ 4.62 *J*_1,2_ 7.0 Hz) as a diagnostic tool as well as from the ^13^C-NMR value for C-1 of GlcN (δ 100.9). The removal of the Troc-protecting group of **49** was achieved with Zn in an Ac_2_O and AcOH mixture to produce protected *N*-acylated hexasaccharide **50**. The removal of benzyl protecting groups in **50** was initially attempted by hydrogenolysis using Pd-C. However, this reaction failed and resulted in side reactions involving the ClAc protecting group leading to an intractable mixture of products. In contrast, significantly improved yields were obtained by deacylation under Zemplén conditions followed by hydrogenolytic cleavage of the benzyl protecting group to produce the desired hexasaccharide **51**. Biotinylation was performed as the usual method to provide target hexasaccharide **E** in 79% yield.

### 2.3. Antigenicity of Oligosaccharides by ELISA

The reactivity of the five oligosaccharides **A**–**E** (NMR spectra provided in the Appendix A). to alveolar echinococcosis (AE) patient sera and cystic echinococcosis (CE) was examined using microplates coated with streptavidin. Contrary to expectations, the antibody response of the CE patient group was not significantly different from that of the normal healthy (NH) group (Figure 7).

Although **A**–**E** display oligosaccharide structures specific to *E. granulosus*, only compounds **A** and **B** showed a modest effect of antigenicity to CE patient serum while no effect was observed with saccharides **C**–**E**. This suggests that the presence of the terminal Galα1-4Gal sequence in oligosaccharides **B**–**E** of *E. granulosus* may suppress a host immune response or the cell surface oligosaccharides on *E. granulosus* may be associated with lower host specificity than *E. multirocularis* [26]. Interestingly, we previously reported that the trisaccharide sequence Galα1-4Galβ1-3GalNAc found on the surface of *E. multilocularis* showed the strongest antigenic response to the AE group among a series of oligosaccharides [8]. Rather unexpected is our finding that oligosaccharides **B**, **D**, and **E** that occur on *E. granulosus* display strong antigenicity to AE patient sera.

## 3. Conclusions

We have developed an efficient synthetic strategy for the preparation of five oligosaccharide-biotin conjugates **A**–**E** which display carbohydrate structures that occur on the surface of *E. granulosus*. The oligosaccharide-biotin conjugates were prepared to study the antigenicity of the compounds to detect antibodies in patient sera infected with *E. granulosus* the cause of CE. Surprisingly, none of the oligosaccharide structures **C**–**E** was able to detect antibodies in sera from patients suffering from CE using our ELISA assay while only a modest response was seen with compounds **A** and **B**. However, glycoconjugates **B**, **D**, and **E** showed good serodiagnostic potential to recognize antibodies in AE patients. Although the oligosaccharide sequence of compound **E** has not been reported in *E. multilocularis*, it showed strong antigenicity to the serum of AE patients. Overall, our results suggest that oligosaccharide-based structures on the cell surface of *E. granulosus* may serve as a diagnostic tool to detect AE. The reasons for this phenomenon are currently not understood. Possible explanations for this observation are: (i) *E. granulosus* induces a suppressed host immune response when compared to *E. multilocularis*; (ii) the presence of Galα1-4Gal terminal capping linkage in *E. granulosus* reduces a host immune response and (iii) oligosaccharide structures present in *E. granulosus* may also be present in *E. multilocularis*. Overall, our investigation encourages future studies in the development of carbohydrate-based antigens as serodiagnostic tools to detect parasitic infections. In particular, our findings that oligosaccharide-biotin probes **D** and **E** can differentiate between sera from AE and CE patients warrant further studies toward the development of serodiagnostic tools to detect parasite-specific infections in humans.

## 4. Experimental

### 4.1. General Procedures

Optical rotations were measured with a Jasco P-1020 digital polarimeter (Tokyo, Japan). ^1^H (500 MHz) and ^13^C-NMR (125 MHz) spectra were recorded with a Varian 500 FT NMR spectrometer (Palo Alto, CA, USA). Me_4_Si and acetone were used as internal standards for CDCl_3_ and D_2_O, respectively. ESI-HRMS was recorded on a JEOL MS T-100 mass spectrometer (Tokyo, Japan). MALDI-TOFMS was recorded on an AB SCIEX Voyager RP mass spectrometer (Framingham, MA, USA). TLC was performed on Silica Gel 60 F254 (E. Merck, Darmstadt, Germany) with detection by the quenching of UV fluorescence and by charring with 10% H_2_SO_4_. Column chromatography was carried out on Silica Gel 60. 5-(Methoxycarbonyl)pentyl 2-azido-4,6-*O*-benzylidene-2-deoxy-α-d-galactopyranoside (**5**) [8], phenyl-2-*O*-benzoyl-3,6-di-*O*-benzyl-4-*O*-chloroacetyl-1-thio-β-d-galactopyranoside (**9**), phenyl 2,3-di-*O*-benzyl-4,6-*O*-di-*tert*-butylsilylene-1-thio-β-d-galactopyranoside (**12**) **[23]**, phenyl 4,6-di-*O*-benzylidene-2*-O*-benzoyl-3-*O*-chloroacetyl-1-thio-β-d-galactopyranoside (**20**) [5], phenyl 2,4,6-tri*-O*-benzyl-3-*O*-chloroacetyl-1-thio-β-d-galactopyranoside (**22**) [25] 2-(trimethylsilyl)ethyl 2-*O*-benzoyl-3,6-di-*O*-benzyl-β-d-galactopyranoside (**32**) [26], 2,6-dimethylphenyl 4,6-di-*O*-acetyl-2,3-di*-O*-benzyl-α-d-galactopyranosyl-(1→4)-2-*O*-benzoyl-3,6-di-*O*-benzyl-β-d-galactopyranosyl-(1→4)-3-*O*-benzoyl-6-*O*-benzyl-2-deoxy-2-(2,2,2-trichloroethoxycarbonylamino)-1-thio-β-d-glucopyranoside (**44**) [11] were prepared as reported. Phenyl 2,3,4,6-tetra-*O*-benzyl-1-thio-β-d-galactopyranoside (**33**) was purchased from Tokyo Chemical Industry Co., Ltd. (TCI), (Tokyo, Japan).

Phenyl 2,4,6-tri-*O*-benzoyl-3-*O*-(2-naphthyl)methyl-1-thio-β-d-galactopyranoside (**3**)

A solution of **1** (5.00 g, 18.4 mmol) and dibutyltin oxide (5.50 g, 22.1 mmol) in 150 mL of dry MeOH was stirred under reflux for 4 h. MeOH was distilled off, the stannylidene derivative was dissolved in toluene (70 mL) and Bu_4_NBr (7.12 g, 22.1 mmol) and NapBr (6.1 g, 27.6 mmol) were added at room temperature. After being stirred for 15 h at 100 °C, the solution was concentrated. To a solution of this residue (**2**) in pyridine (4 mL) benzoyl chloride (60 mL, 82.8 mmol) was added, and the reaction mixture was stirred for 5 h at room temperature. Toluene was added and evaporated, then the residue was dissolved in CHCl_3_, washed with 5% HCl, aq NaHCO_3_ and water, dried (MgSO_4_), and concentrated. The product was purified by silica gel column chromatography using 8:1 hexane-EtOAc as eluent to give **3** (8.46 g, 64% 2 steps). ${\left[\alpha \right]}_{D}^{22}$ + 66.0 (*c* =1.0, CHCl_3_). ^1^H-NMR (500 MHz, CDCl_3_): δ 8.01–7.16 (m, 27H, 4×Ph, naphtyl), 5.97 (d, 1H, *J*_3, 4_ = 2.5 Hz, H-4), 5.55 (t, 1H, *J*_1, 2_ = *J*_2, 3_ = 9.8 Hz, H-2), 4.83 and 4.64 (each d, 2H, *J*_gem_ = 12.8 Hz, naphtylmethylene), 4.82 (d, 1H, H-1), 4.58 (dd, 1H, *J*_5, 6a_ = 7.4 Hz, *J*_6a, 6b_ = 11.6 Hz, H-6a), 4.50–4.47 (dd, 1H, *J*_5, 6b_ = 5.0 Hz, H-6b), 4.15 (br.t, 1H, H-5), 3.88 (dd, 1H, H-3). ^13^C-NMR (125 MHz, CDCl_3_): δ 166.1, 165.8, 165.1, 134.5, 133.43, 133.41, 133.3, 133.1, 132.96, 132.93, 132.0, 130.1, 129.9, 129.8, 129.5, 129.2, 128.7, 128.5, 128.44, 128.36, 128.2, 128.0, 127.8, 127.6, 126.9, 126.0, 125.89, 125.86, 86.0 (C-1), 77.2 (C-3), 75.2 (C-5), 71.0 (naphtylmetylene), 69.4 (C-2), 66.8 (C-4), 63.1 (C-6). HR-ESIMS: calcd for C_44_H_36_O_8_SNa: *m/z* 747.2029; found: *m/z* 747.2003 [M + Na]^+^.

Phenyl 2,4,6-tri-*O*-benzoyl-3-*O*-chloroacetyl-1-thio-β-d-galactopyranoside (**4**)

A solution of **3** (8.46 g, 11.07 mmol) in CH_2_Cl_2_—H_2_O (20:1, 84 mL) was treated with DDQ (5.31 g, 23.4 mmol) at room temperature and then was stirred for 6 h. After concentration, the residue was added to the water, extracted with CHCl_3_, and the organic layer was proceeded as usual. The product was purified by silica gel column chromatography (3:1 hexane-EtOAc) as eluent to give intermediate (6.19 g, 96%). [MALDI-TOFMS: calcd for C_33_H_28_O_8_SNa, *m/z* 607.1; found, *m/z* 607.6 [M + Na]^+^]. To a solution of this compound (4.00 g, 6.84 mmol) in CH_2_Cl_2_/pyridine (15:1, 64 mL) was added chloroacetyl chloride (ClAcCl ) (817 μL, 10.3 mmol), and the reaction mixture was stirred for 1 h at 0°C. The residue was dissolved in CHCl_3_, washed with 5% HCl, aq NaHCO_3_ and water, dried (MgSO_4_), and concentrated. The product was purified by silica gel column chromatography using 2:1 hexane-EtOAc as eluent to give **4** (3.9 g, 83% 2 steps). ${\left[\alpha \right]}_{D}^{22}$ + 32.9 (*c* =1.0, CHCl_3_). ^1^H-NMR (500 MHz, CDCl_3_): δ 8.03–7.24 (m, 20H, 4×Ph), 5.85 (d, 1H, *J*_3, 4_ = 3.2 Hz, H-4), 5.59 (d, 1H, *J*_1, 2_ = *J*_2, 3_ = 9.8 Hz, H-2), 5.45 (dd, 1H, H-3), 4.97 (d, 1H, H-1), 4.64 (dd, 1H, *J*_5, 6a_ = 7.0 Hz, *J*
_6a, 6b_ = 11.4 Hz, H-6a), 4.40 (dd, 1H, *J*_5, 6b_ = 5.8 Hz, H-6b), 4.32 (t, 1H, H-5), 3.88 and 3.83 (each d, 1H, *J*_gem_ = 15.2 Hz, CH_2_Cl). ^13^C-NMR (125 MHz, CDCl_3_): δ 166.7, 166.0, 165.7, 165.1, 134.0, 133.8, 133.6, 133.4, 133.1, 133.0, 131.0, 130.0, 129.9, 129.8, 129.3, 129.1, 128.9, 128.7, 128.6, 128.5, 85.9 (C-1), 74.9 (C-5), 74.1 (C-3), 67.9 (C-4), 67.6 (C-2), 62.3 (C-6), 40.4 (CH_2_Cl). HR-ESIMS: calcd for C_35_H_29_ClO_9_SNa: *m/z* 683.1119; found: *m/z* 683.1142 [M + Na]^+^.

5-(Methoxycarbonyl)pentyl 2,4,6-tri-*O*-benzoyl-3-*O*-(2-naphthyl)methyl-β-d-galacto-pyranosyl-(1→3)-2-azido-4,6-*O*-benzylidene-2-deoxy-α-d-galactopyranoside (**6**)

To a solution of **5** (100 mg, 0.237 mmol) and **3** (223 mg, 0.308 mmol) in dry CH_2_Cl_2_ (2 mL) powdered AW300 (300 mg) was added, and the mixture was stirred under Ar atmosphere for 2 h at room temperature, then cooled to −40 °C. NIS (139 mg, 0.62 mmol) and TfOH (5.5 μL, 0.06 mmol) were added to the mixture, which was stirred for 0.5 h at –40 °C, then neutralized with Et_3_N. The precipitates were filtered off and washed with CHCl_3_. The combined filtrate and washings were successively washed with saturated aqueous Na_2_S_2_O_3_ and water, dried (MgSO_4_), and concentrated. The product was purified by silica gel column chromatography (70:1 toluene-acetone) to give **6** (175 mg, 71%). ${\left[\alpha \right]}_{D}^{23}$ + 103.1 (*c* =1.0, CHCl_3_). ^1^H-NMR (500 MHz, CDCl_3_): δ 8.19–6.85 (m, 27H, 4×Ph, naphtylmethylene), 5.96 (d, 1H, *J*_3′, 4′_ = 2.4 Hz, H-4′), 5.68 (dd, 1H, *J*_1′, 2′_ = 8.1 Hz, *J*_2′, 3′_ = 9.9 Hz, H-2′), 5.46 (s, 1H, PhCH), 4.53 (d, 1H, H-1′), 4.89 (d, 1H, *J*_1, 2_ = 3.3 Hz, H-1), 4.87–4.80 (m, 1H, naphtylmethylene), 4.72–4.68 (m, 1H, H-6′a), 4.68—4.65 (m, 1H, naphtylmethylene), 4.45–4.43 (m, 1H, H-6′b), 4.42 (d, 1H, *J*_3, 4_ = 2.6 Hz, H-4), 4.16 (t, 1H, H-5′), 4.11–4.07 (m, 1H, H-6a), 4.03 (dd, 1H, *J*_2, 3_ = 10.9 Hz, H-3), 3.88 (dd, 1H, *J*_3′ 4′_ = 3.3 Hz, H-3′), 3.72–3.68 (m, 2H, H-2, H-6b), 3.66—3.59 (m, 4H, -OCH_2_-_,_ -OCH_3_), 3.43 (s, 1H, H-5), 3.41–3.38 (m, 1H, -OCH_2_-), 2.30–2.26 (m, 2H, -CH_2_-), 1.64–1.53 (m, 4H, 2×-CH_2_-), 1.37–1.25 (m, 2H, -CH_2_-). ^13^C-NMR (125 MHz, CDCl_3_): δ 174.0, 166.0, 165.2, 137.7, 134.5, 133.46, 133.43, 132.94, 132.91, 130.4, 130.13, 130.08, 129.9, 129.8, 129.7, 129.6, 129.3 129.1, 128.7, 128.6, 128.5, 128.4, 128.19, 128.16. 128.05, 127.79, 127.57, 126.8, 126.1, 126.0, 125.9, 125.8, 102.9 (C-1′), 100.6 (PhCH), 98.6 (C-1), 76.2 (C-3′), 76.1 (C-3), 75.9 (C-4), 71.4 (C-5′),71.0 (naphtylmethylene), 70.96 (C-2′), 68.95 (C-6), 68.2(OCH_2_), 66.6 (C-4′), 63.0 (C-5, C-6′), 58.4 (C-2), 51.4 (OCH_3_), 33.8, 28.9, 25.6, 24.6. HR-ESIMS: calcd for C_58_H_57_N_3_O_15_Na: *m*/*z* 1058.3687; found: *m*/*z* 1058.3667 [M + Na]^+^.

5-(Methoxycarbonyl)pentyl 2,4,6-tri-*O*-benzoyl-3-*O*-chloroacetyl-β-d-galactopyranosyl-(1→3)-2-azido-4,6-*O*-benzylidene-2-deoxy-α-d-galactopyranoside (**7**)

To a solution of **5** (200 mg, 0.48 mmol) and **4** (377 mg, 0.57 mmol) in dry CH_2_Cl_2_ (5 mL) was added powdered MS AW300 (600 mg), and the mixture was stirred under Ar atmosphere for 2 h at room temperature, then cooled to −40 °C. NIS (192 mg, 0.85 mmol) and TfOH (15 μL, 0.17 mmol) were added to the mixture, which was stirred for 0.5 h at −40 °C, then neutralized with Et_3_N. The precipitates were filtered off and washed with CHCl_3_. The combined filtrate and washings were successively washed with saturated aqueous Na_2_S_2_O_3_ and water, dried (MgSO_4_), and concentrated. The product was purified by silica gel column chromatography (15:1 toluene-EtOAc) to give **7** (413 mg, 89%). ${\left[\alpha \right]}_{D}^{22}$ + 96.3 (*c* =1.0, CHCl_3_). ^1^H-NMR (500 MHz, CDCl_3_): δ 8.13–7.15 (m, 20H, 4×Ph), 5.85 (d, 1H, *J*_3′, 4′_ = 2.6 Hz, H-4′), 5.73 (br.t, 1H, H-2′), 5.49 (s, 1H, PhCH), 5.46 (dd, 1H, *J*_2′, 3′_ = 10.5 Hz, H-3′), 5.11 (d, 1H, *J*_1′, 2′_ = 7.9 Hz, H-1′), 4.92 (d, 1H, *J*_1, 2_ = 2.8 Hz, H-1), 4.78–4.73 (m, 1H, H-6′a), 4.44 (d, 1H, *J*_3, 4_ = 1.8 Hz, H-4), 4.40–4.34 (m, 2H, H-5′, H-6′b), 4.14–4.12 (m, 2H, H-3, H-6a), 3.91 and 3.85 (each d, *J*_gem_ = 15.2 Hz, 2H, CH_2_Cl), 3.76–3.71 (m, 2H, H-2, H-6b), 3.67–3.63 (m, 4H, -OCH_2_-, -OCH_3_), 3.49 (s, 1H, H-5), 3.46–3.42 (m, 1H, -OCH_2_-), 2.35–2.30 (m, 2H, -CH_2_-), 1.68–1.58 (m, 4H, 2×-CH_2_-), 1.41–1.26 (m, 2H, -CH_2_-). ^13^C-NMR (125 MHz, CDCl_3_): δ 174.0, 166.8, 165.84, 165.82, 165.0, 137.7, 133.8, 133.5, 133.3, 130.0, 129.7, 129.6, 129.3, 129.27, 129.2, 129.0, 128.98, 128.76, 128.70, 128.65, 128.5, 128.3, 128.2, 128.1, 126.1, 125.2, 102.8 (C-1′), 100.6 (PhCH), 98.5 (C-1), 76.3 (C-3), 75.8 (C-4), 73.0 (C-3′), 71.2 (C-5′), 69.2 (C-2′), 69. 0 (C-6), 68.3 (-OCH_2_-), 67.7 (C-4′), 63.0 (C-5), 62.2 (C-6′), 58.4 (C-2), 51.5 (OCH_3_), 40.4 (-CH_2_Cl), 33.8, 29.6, 28.9, 25.6, 24.6, 21.4. HR-ESIMS: calcd for C_49_H_50_ClN_3_O_16_Na: *m*/*z* 994.2777; found: *m*/*z* 994.2764 [M + Na]^+^.

5-(Methoxycarbonyl)pentyl 2,4,6-tri-*O*-benzoyl-β-d-galactopyranosyl-(1→3)-2-azido-4,6-*O*-benzylidene-2-deoxy-α-d-galactopyranoside (**8**)

(I) A solution of **6** (175 mg, 0.17 mmol) in CH_2_Cl_2_-H_2_O (19:1, 2 mL) was treated with DDQ (96 mg, 0.42 mmol) at room temperature and then was stirred for 6 h. After concentration, the residue was added to the water, extracted with CHCl_3_, and the organic layer was proceeded as usual. The product was purified by silica gel column chromatography (2.5:1 hexane-AcOEt) as eluent to give **8** (94.5 mg, 63%).

(II) A solution of **7** (720 mg, 0.74 mmol) in MeOH-Pyr. (3:1, 8 mL) was treated with thiourea (169 mg, 2.22 mmol) under reflux for 2 h. After concentration, the residue was added to the water, extracted with CHCl_3_, and the organic layer was proceeded as usual. The product was purified by silica gel column chromatography (18:1 toluene-acetone) as eluent to give **8** (558 mg, 84%). ${\left[\alpha \right]}_{D}^{24}$ + 44.2 (*c* =1.0, CHCl_3_). ^1^H-NMR (500 MHz, CDCl_3_): δ 8.15–7.26 (m, 20H, 4×Ph), 5.75 (d, 1H, *J*_3′, 4′_ = 3.1 Hz, H-4′), 5.46 (dd, 1H, *J*_1′, 2′_ = 7.8 Hz, *J*_2′, 3′_ = 10 Hz, H-2′), 5.44 (s, 1H, PhCH), 5.05 (d, 1H, H-1′), 4.95 (d, 1H, *J*_1, 2_ = 3.5 Hz, H-1), 4.69 (dd, 1H, *J*_5′, 6′a_= 7.4 Hz, *J*_6′a, 6′b_ = 11.5 Hz, H-6′a), 4.44–4.41 (m, 2H, H-4, H-6′b), 4.23–4.21 (m, 1H, H-5′), 4.16–4.08 (m, 3H, H-3, H-6a, H-3′), 3.77 (dd, *J*_2, 3_ = 10.7 Hz, 1H, H-2), 3.71–3.63 (m, 5H, H-6b, -OCH_2,_ CH_3_), 3.48–3.45 (m, 1H, -OCH_2,_-), 3.43 (s, 1H, H-5), 3.00 (d, 1H, OH), 2.34–2.31 (m, 2H, -CH_2_-), 1.69–1.59 (m, 4H, 2×-CH_2_-), 1.41–1.26 (m, 2H, -CH_2_-). ^13^C-NMR (125 MHz, CDCl_3_): δ 174.0, 167.3, 166.2, 166.0, 137.7, 133.6, 133.45, 133.43, 130.1, 129.9, 129.7, 129.6, 129.4, 129.1, 128.8, 128.7, 128.5, 128.3, 128.1, 126.1, 102.6 (C-1′), 100.7 (PhCH), 98.6 (C-1), 76.3 (C-3), 76.0 (C-4), 73.7 (C-2′), 72.2 (C-3′), 71.6 (C-5′), 70.5 (C-4′), 69.0 (C-6), 68.3 (OCH_2_), 63.0 (C-5), 62.8 (C-6′), 58.6 (C-2), 51.5 (OCH_3_), 33.9, 29.6, 29.0, 25.6, 24.7. HR-ESIMS: calcd for C_47_H_49_N_3_O_15_Na: *m/z* 918.3061; found: *m/z* 918.3045 [M + Na]^+^.

5-(Methoxycarbonyl)pentyl 2-*O*-benzoyl-3,6-di-*O*-benzyl-4-*O*-chloroacetyl-β-d-galacto-pyranosyl-(1→3)-2,4,6-tri-*O*-benzoyl-β-d-galactopyranosyl-(1→3)-2-azido–4,6-*O*-benzylidene-2-deoxy-α-d-galactopyranoside (**10**)

Compound **10** was prepared from **8** (648 mg, 0.72 mmol) and **9** (554 mg, 0.87 mmol) as described for preparation of **7**. The product was purified by silica gel column chromatography (30:1 toluene-acetone) to give **10** (802 mg, 78%). ${\left[\alpha \right]}_{D}^{24}$ + 53.3 (*c* =1.0, CHCl_3_). ^1^H-NMR (500 MHz, CDCl_3_): δ 8.10–6.96 (m, 35H, 7×Ph), 5.74 (d, 1H, *J*_3′, 4′_ = 3.6 Hz, H-4′), 5.57 (dd, 1H, *J*_1′, 2′_ = 8.1 Hz, *J*_2′, 3′_ = 9.9 Hz, H-2′), 5.52 (d, 1H, *J*_3′, 4′_ = 2.8 Hz, H-4″), 5.34 (s, 1H, PhCH), 5.11 (dd, 1H, *J*_1′, 2′_ = 7.9 Hz, *J*_2′, 3′_ = 10.2 Hz, H-2″), 4.89 (d, 1H, H-1′), 4.83 (d, 1H, *J*_1, 2_ = 3.3 Hz, H-1), 4.69 (d, 1H, H-1″), 4.58–4.13 (m, 7H, H-4, H-5′, H-6′a, b, H-3″, 2×PhCH_2_), 4.02 (br. dd, 2H, H-6″a, b), 3.93 (dd, 1H, *J*_2, 3_ = 11.0 Hz, *J*_3, 4_ = 3.0 Hz, H-3), 3.87–3.29 (m, 13H, H-2, H-3, H-5, H-6a, b, H-3′, -CH_2_Cl, -OCH_2_-, -OCH_3_), 2.34–2.31 (m, 2H, -CH_2_-), 1.69–1.59 (m, 4H, -CH_2_-), 1.41–1.26 (m, 2H, -CH_2_-). ^13^C-NMR (125 MHz, CDCl_3_): δ 174.0, 166.8, 166.3, 166.0, 164.5, 164.4, 137.6, 137.4, 136.8, 133.3, 133.1, 132.6, 132.5, 130.2, 129.81, 129.77, 129.63, 129.60, 129.4, 128.57, 128.50, 128.47, 128.2, 128.1, 128.03, 127.98, 127.95, 127.88, 127.6, 126.0, 102.8 (C-1 of Gal a), 101.1 (C-1 of Gal b), 100.4, 98.6 (C-1 of GalN), 77.2, 75.8, 75.7, 75.6, 73.7, 72.0, 71.9, 71.3, 70.8, 70.7, 70.5, 68.8, 68.2, 67.6, 67.1, 63.4, 63.0, 58.3, 51.5, 40.5, 33.9, 29.7, 28.9, 25.6, 24.6. HR-ESIMS: calcd for C_76_H_76_ClN_3_O_22_Na: *m/z* 1440.4507; found: *m/z* 1440.4543 [M + Na]^+^.

5-(Methoxycarbonyl)pentyl 2-*O*-benzoyl-3,6-di-*O*-benzyl-β-d-galactopyranosyl-(1→3)-2,4,6-tri-*O*-benzoyl-β-d-galactopyranosyl-(1→3)-2-azido-4,6-*O*-benzylidene-2-deoxy-α-d-galacto-pyranoside (**11**)

Compound **11** was prepared from **10** (77.5 mg, 54.6 μmol) as described for preparation of **8**, yielding 62.4 mg (85%). ${\left[\alpha \right]}_{D}^{24}$ + 43.3 (*c* =1.0, CHCl_3_). ^1^H-NMR (500 MHz, CDCl_3_): δ 8.07–6.99 (m, 35H, 7×Ph), 5.79 (d, 1H, *J*_3′, 4′_ = 3.6 Hz, H-4′), 5.58 (dd, 1H, *J*_1′, 2′_ = 8.0 Hz, *J*_2′, 3′_ = 10.0 Hz, H-2′), 5.52 (d, 1H, *J*_3′, 4′_ = 2.8 Hz, H-4“), 5.35 (s, 1H, PhCH), 5.21 (dd, 1H, *J*_1′, 2′_ = 7.8 Hz, *J*_2′, 3′_ = 9.6 Hz, H-2″), 4.89 (d, 1H, H-1′), 4.84 (d, 1H, *J*_1, 2_ = 3.5 Hz, H-1), 4.66 (d, 1H, H-1″), 4.60–3.94 (m, 10H, H-3, H-4, H-5′, H-6′a, b, H-3″, H-6″a, b, 2×PhCH_2_), 3.77–3.32 (m, 11H, H-2, H-3, H-5, H-6a, b, H-3′, -OCH_2_, OCH_3_), 2.34–2.31 (m, 2H, -CH_2_-), 1.69–1.59 (m, 4H, -2×CH_2_-), 1.41–1.26 (m, 2H, -CH_2_-). ^13^C-NMR (125 MHz, CDCl_3_): δ 174.0, 166.1, 165.9, 164.7, 164.4, 138.1, 137.6, 137.0, 133.3, 133.1, 132.7, 132.4, 130.2, 129.8, 129.70, 129.67, 129.60,129.56, 128.56, 128.50, 128.48, 128.40, 128.3, 128.1, 127.9, 127.79, 127.75, 127.6, 126.0, 102.8 (C-1 of Gal a), 101.1 (C-1 of Gal b), 100.4, 98.6 (C-1 of GalN), 78.0, 76.0, 75.8, 75.6, 73.7, 72.0, 71.5, 71.05, 70.95, 70.6, 68.88, 68.85, 68.2, 65.9, 63.2, 63.0, 58.3, 51.5, 33.9, 29.0, 25.6, 24.6. HR-ESIMS: calcd for C_74_H_75_N_3_O_21_Na: *m/z* 1364.4791; found: *m/z* 1364.4755 [M + Na]^+^.

5-(Methoxycarbonyl)pentyl 2,3-di-*O*-benzyl-4,6-*O*-di-*tert*-butylsilylene-α-d-galactopyranosyl-(1→4)-2-*O*-benzoyl-3,6-di-*O*-benzyl-β-d-galactopyranosyl-(1→3)–2,4,6-tri-*O*-benzoyl-β-d-galactopyranosyl-(1→3)-2-azido-4,6-*O*-benzylidene–2-deoxy-α-d-galactopyranoside (**13**)

Compound **13** was prepared from **11** (60 mg, 44.7 μmol) and **12** (39.8 mg, 67.1 μmol) as described for preparation of **6**. The product was purified by silica gel column chromatography (10:1 toluene-ethyl acetate) to give **13** (69.7 mg, 85%). ${\left[\alpha \right]}_{D}^{24}$ + 71.2 (*c* =1.0, CHCl_3_). ^1^H-NMR (500 MHz, CDCl_3_): δ 8.02–6.95 (m, 45H, 9×Ph), 5.81 (d, 1H, *J*_3′, 4′_ = 3.3 Hz, H-4′), 5.61 (dd, 1H, *J*_1′, 2′_ = 7.9 Hz, *J*_2′, 3′_ = 9.7 Hz, H-2′), 5.33 (s, 1H, PhCH), 5.26 (dd, 1H, *J*_1′, 2′_ = 7.9 Hz, *J*_2′, 3′_ = 10.2 Hz, H-2″), 4.88 (d, 1H, *J*_1, 2_ = 6.7 Hz, H-1 of Gal a), 4.86 (d, 1H, *J*_1, 2_ = 2.6 Hz, H-1 of Gal c), 4.84 (d, 1H, *J*_1, 2_ = 3.0 Hz, H-1 of GalN), 4.67 (d, 1H, *J*_1, 2_ = 8.0 Hz, H-1 of Gal b), 0.97 and 0.95 (each s, 18H, 2×C(CH_3_)_3_. ^13^C-NMR (125 MHz, CDCl_3_): δ 174.0, 166.0, 164.7, 164.5, 139.22, 139.19, 138.2, 137.6, 137.4, 133.2, 132.9, 132.7, 132.3, 130.1, 129.8, 129.6, 129.5, 128.5, 128.4, 128.3, 128.2, 128.1, 128.0, 127.78, 127.73, 127.67, 127.3, 127.2, 127.1, 126.0, 102.8 (C-1 of Gal a), 101.1 (C-1 of Gal b), 100.5 (C-1 of Gal c), 100.4, 98.6 (C-1 of GalN), 78.6, 78.1, 75.8, 75.7, 75.6, 74.5, 74.1, 73.4, 73.1, 72.1, 71.6, 71.4, 71.1, 70.9, 70.7, 70.5, 68.8, 68.2, 67.7, 67.0, 63.3, 63.0, 58.3, 51.5, 33.8, 29.7, 29.0, 27.6, 27.4, 25.6, 24.6, 23.2, 20.7. HR-ESIMS: calcd for C_102_H_113_N_3_O_26_SiNa: *m/z* 1846.7279; found: *m/z* 1846.7245 [M + Na]^+^.

5-(Methoxycarbonyl)pentyl 3,4,6-tri-*O*-acetyl-2-*O*-benzoyl-β-d-galactopyranosyl-(1→3)-2,4,6-tri-*O*-benzoyl-β-d-galactopyranosyl-(1→3)-2-acetamido-4,6-di-*O*-acetyl-2-deoxy-α-d-galactopyranoside (**14**)

To a solution of **11** (137 mg, 0.10 mmol) in MeOH—THF—NH_3_aq. (3:1:0.1, 4.1 mL) Pd/C (150 mg) was added. The mixture was stirred for 80 min at room temperature under H_2_ atmosphere. After completion of the reaction, the mixture was filtered through Celite. The filtrate was concentrated with toluene. To a solution of this compound in AcOH (2.0 mL) was hydrogenolysed in the presence of Pd/C (150 mg) for 18 h at room temperature. The mixture was filtered and concentrated, and the residue was acetylated with acetic anhydride (3.0 mL) in pyridine (5.0 mL). After concentration, the residue underwent de-benzylidenation with TsOH (30 mg) in CHCl_3_-MeOH (2:1, 6 mL) for 15 h at room temperature, and was then neutralized with Et_3_N. After concentration, the residue was acetylated with acetic anhydride (1.0 mL) in pyridine (1.5 mL) for 5 h at room temperature. After the reaction, toluene was added and co-evaporated several times. The product was purified by silica gel column chromatography (5:1 CHCl_3_-MeOH) to give **14** (68 mg, 51%, 5 steps). ${\left[\alpha \right]}_{D}^{25}$ + 69.2 (*c*=1.0, CHCl_3_). ^1^H-NMR (500 MHz, CDCl_3_): δ 8.10–7.17 (m, 20H, 4×Ph), 5.83 (d, 1H, *J*_3′, 4′_ = 3.1 Hz, H-4′), 5.49–5.46 (m 2H, H-4, 2′), 5.38 (d, 1H, *J* = 8.3 Hz, NH), 5.27 (d, 1H, *J*_3″, 4″_ = 2.4 Hz, H-4″), 5.23 (dd, 1H, *J*_1′, 2′_ = 7.8 Hz, *J*_2′, 3′_ = 10.5 Hz, H-2″), 4.77 (d, 1H, *J*_1″, 2″_ = 7.6 Hz, H-1″), 4.82 (d, 1H, *J*_1, 2_ = 3.5 Hz, H-1 of Gal N), 4.81 (d, 1H, *J*_1, 2_ = 8.1 Hz, H-1of Gal’). 4.50–3.23 (m 14H, H-2, 3, 5, 6a, 6b, 2′, 5′, 6′a, 6′b, 5″, 6″a, 6″b, -OCH_2_-), 3.67 (s, 3H, OMe), 2.26–2.23 (m, 2H, -CH_2_-), 2.05, 2.002, 1.999, 1.989 (each s, 12H, 4×Ac), 1.74–1.72 (m, 4H, -CH_2_-), 1.43–1.21 (m, 2H, -CH_2_-). ^13^C-NMR (125 MHz, CDCl_3_): δ 174.0, 170.4, 170.3, 170.2, 170.0, 169.9, 169.7, 169.2, 165.7, 164.3, 133.2, 133.13, 133.08, 132.7, 130.1, 129.7, 129.5, 129.44, 129.37, 129.1, 128.9, 128.42, 128.37, 128.35, 128.0, 101.0 (C-1″), 100.8 (C-1′), 97.1 (C-1), 76.7, 74.7, 72.1, 71.9, 70.7, 70.5, 70.1, 69.7, 69.2, 68.6, 67.7, 67.1, 63.0, 62.7, 60.9, 51.5, 48.9, 33.7, 28.5, 25.6, 24.64, 22.3, 20.7, 20.63, 20.59, 20.4, 20.3. HR-ESIMS: calcd for C_65_H_73_NO_27_Na: *m/z* 1322.4268; found: *m/z* 1322.4249 [M + Na]^+^.

Biotinylated trisaccharide (**A**)

To a solution of **14** (68 mg, 52.3 μmol) in MeOH (1.0 mL) NaOMe (30 mg) was added and the mixture was stirred at 40 °C for 2 h, then neutralized with Amberlite IR 120 [H^+^]. The mixture was filtered off and concentrated. The product was purified by Sephadex LH-20 column chromatography in H_2_O to give **15** (38 mg, quant.) [MALDI-TOFMS: calcd for C_27_H_47_NO_18_Na, *m*/*z* 696.3; found, *m*/*z* 696.7 [M + Na]^+^]. The residue (26 mg, 37 μmol) was dissolved in anhydrous ethylenediamine (5 mL) and heated at 70 °C for 48 h. The mixture was concentrated with toluene and the product was purified by Sephadex LH-20 column chromatography in H_2_O to give an amine intermediate. The amine was dissolved in DMF (4.0 mL), and the pH was adjusted to 8–9 using DIPEA. Biotine-NHS (15.2 mg, 45.0 μmol) was added and the reaction stirred for 12 h at room temperature. Toluene was added to and evaporated from the residue several times. The product was purified by Sephadex LH-20 column chromatography in H_2_O to give **A** (23.0 mg, 67%). ${\left[\alpha \right]}_{D}^{25}$ + 82.4 (*c*=0.4, CHCl_3_). ^1^H-NMR (500 MHz, D_2_O): δ 4.70 (d, 1H, *J*_1, 2_ = 4.0 Hz, H-1), 4.42 (d, 1H, *J*_1, 2_ = 7.0 Hz, H-1″), 4.34 (d, 1H, *J*_1, 2_ = 7.5 Hz, H-1′). ^13^C-NMR (125 MHz, CDCl_3_): δ 176.8, 176.6, 174.0, 164.9, 104.0 (C-1″), 103.9 (C-1′), 96.6 (C-1), 81.6, 77.2, 74.6, 74.2, 72.1, 70.6, 70.1, 69.4, 68.2, 68.0, 67.4, 61.6, 60.8, 60.5, 59.8, 54.9, 48.3, 39.3, 38.2, 38.1, 35.4, 35.1, 27.8, 27.5, 27.3, 24.70, 24.69, 24.5, 21.6, 20.7, 20.63, 20.59, 20.4, 20.3. HR-ESIMS: calcd for C_38_H_65_N_5_O_19_SNa: *m/z* 950.3892.; found: *m/z* 950.3984 [M + Na]^+^.

5-(Methoxycarbonyl)pentyl 2,3-di-*O*-benzyl-4,6-di-*O*-acetyl-α-d-galactopyranosyl-(1→4)-2-*O*-benzoyl-3,6-di-*O*-benzyl-β-d-galactopyranosyl-(1→3)-2,4,6-tri-*O*-benzoyl-β-d-galactopyranosyl-(1→3)-2-azido-4,6-di-*O*-acetyl-2-deoxy-α-d-galactopyranoside (**16**)

A solution of **13** (217 mg, 0.12 mmol) in Pyr. (2.5 mL) was added HF—Pyr. (1.0 mL) at 0 °C and then was stirred for 4 h. The reaction mixture was added to water, extracted with ethyl acetate, and the organic layer was washed with saturated aqueous NaHCO_3_ and water, dried (MgSO_4_), and concentrated. A solution of the residue in 80% AcOH (4 mL) was stirred at 70 °C for 5 h, then was diluted with toluene and concentrated. The residue was treated with Ac_2_O (1.1 mL) in pyridine (2 mL). The reaction mixture was poured into ice-water and extracted with CHCl_3_. The extract was washed sequentially with 5% HCl, saturated aqueous NaHCO_3_ and water, dried (MgSO_4_), and concentrated. The product was purified by silica gel column chromatography (10:1 toluene-acetone) to give **16** (136 mg, 65%, 3 steps). ${\left[\alpha \right]}_{D}^{24}$ +66.8 (*c* = 0.50, CHCl_3_). ^1^H-NMR (500 MHz, CDCl_3_): δ 8.06–6.94 (m, 40H, 8×Ph), 5.84 (d, 1H, *J*_3,4_ = 3.1 Hz, H-4 of Gal b), 5.55–5.49 (m 4H, H-4 of GalN, H-2, 4 of Gal a, H-4 of Gal c), 5.30 (dd, 1H, *J*_1,2_ = 7.5 Hz, *J*_2,3_ = 10.0 Hz, H-2 of Gal b), 4.97 (d, 1H, *J*_1, 2_ = 3.5 Hz, H-1 of Gal c), 4.81 (d, 1H, *J*_1, 2_ = 7.0 Hz, H-1 of Gal a), 4.80 (d, 1H, *J*_1, 2_ = 3.5 Hz, H-1 of GalN), 4.67 (d, 1H, *J*_1, 2_ = 7.0 Hz, H-1 of Gal b), 3.65 (s, 3H, OMe), 2.36–2.31 (m, 2H, -CH_2_-), 2.02×2, 1.98, 1.85 (each s, 12H, 4×Ac), 1.66–1.54 (m, 4H, -CH_2_-), 1.35–1.25 (m, 2H, -CH_2_-). ^13^C-NMR (125 MHz, CDCl_3_): δ 173.9, 170.5, 169.5, 166.1, 165.6, 164.5, 138.9, 138.3, 137.2, 133.0, 132.8, 130.0, 129.9, 129.7, 129.5, 128.4, 128.3, 128.2, 128.2, 128.1, 127.9, 127.9, 127.7, 127.6, 127.5, 127.3, 101.9 (C-1 of Gal a), 101.3 (C-1 of Gal b), 100.5 (C-1 of Gal c), 97.9 (C-1 of GalN), 76.4, 75.8, 74.6, 73.9, 73.7, 73.0, 72.0, 71.9, 71.6, 71.5, 69.8, 68.1, 67.8, 67.2, 67.0, 62.8, 61.5, 59.1, 51.5, 33.9, 29.7, 28.9, 25.6, 24.5, 20.8, 20.7, 20.6. HR-ESIMS: calcd for C_95_H_101_N_3_O_30_Na: *m*/*z* 1786.6368.; found: *m*/*z* 1786.6384 [M + Na]^+^.

5-(Methoxycarbonyl)pentyl 2,3,4,6-tetra-*O*-acetyl-α-d-galactopyranosyl-(1→4)-3,6-di-*O*-acetyl-2-*O*-benzoyl-β-d-galactopyranosyl-(1→3)-2,4,6-tri-*O*-benzoyl-β-d-galactopyranosyl-(1→3)-2-acetamido-4,6-di-*O*-acetyl-2-deoxy-α-d-galactopyranoside (**17**)

To a solution of **16** (300 mg, 0.17 mmol) in THF—H_2_O (6:1, 7.0 mL) was added PPh_3_ (50 mg, 0.19 mmol). The mixture was stirred under reflux for 3.5 h after completion of the reaction, the reaction mixture was added to water, extracted with ethyl acetate, and the organic layer was washed with saturated aqueous NaHCO_3_ and water, dried (MgSO_4_), and concentrated. The residue was acetylated with acetic anhydride (4.0 mL) in pyridine (6.0 mL). After the reaction was quenched with MeOH, the residue was diluted with toluene and concentrated. The product was purified by silica gel column chromatography (10:1 toluene-acetone) to give acetamide compound. This compound in THF-MeOH (1:1, 2.0 mL) was hydrogenolysed in the presence of Pd(OH)_2_/C (75 mg) for 19 h at room temperature, and the mixture was filtered and concentrated. The residue was acetylated with acetic anhydride (1.0 mL) in pyridine (1.0 mL). After the reaction was quenched with MeOH, toluene was added and co-evaporated several times. The product was purified by silica gel column chromatography (10:1 toluene-acetone) to give **17** (175 mg, 65%, 4 steps). ${\left[\alpha \right]}_{D}^{24}$ +91.6 (*c* = 0.59, CHCl_3_). ^1^H-NMR (500 MHz, CDCl_3_): δ 8.14–7.16 (m, 20H, 4×Ph), 5.90 (d, 1H, *J*_3,4_ = 3.1 Hz, H-4 of Gal a), 5.51–5.13 (m 8H, H-4 of GalN, H-2 of Gal a, H-2,3 of Gal b, H-2,3,4 of Gal c, NH), 4.99 (d, 1H, *J*_1, 2_ = 3.5 Hz, H-1 of Gal c), 4.86 (d, 1H, *J*_1, 2_ = 3.5 Hz, H-1 of GalN), 4.80 (d, 1H, *J*_1, 2_ = 7.2 Hz, H-1 of Gal a), 4.79 (d, 1H, *J*_1, 2_ = 7.8 Hz, H-1 of Gal b), 3.67 (s, 3H, OMe), 2.36–2.25 (m, 2H, -CH_2_-), 2.18, 2.11, 2.06, 2.04, 2.02, 2.00 × 2, 1.94, 1.91 (each s, 27H, 9×Ac), 1.68–1.56 (m, 4H, -CH_2_-), 1.35–1.25 (m, 2H, -CH_2_-). ^13^C-NMR (125 MHz, CDCl_3_): δ 175.6, 174.0, 170.6, 170.5, 170.4, 170.07, 169.92, 169.8, 169.3, 166.2, 165.3, 164.5, 164.1, 137.9, 133.1, 133.0, 132.7, 131.0, 130.2, 129.8, 129.7, 129.5, 129.3, 129.2, 129.01, 128.96, 128.4, 128.2, 128.1, 125.3, 101.0 (C-1 of Gal a), 100.6 (C-1 of Gal b), 98.6 (C-1 of Gal c), 97.1 (C-1 of GalN), 75.6, 74.4, 72.10, 71.95, 69.6, 69.3, 68.6, 68.4, 68.1, 67.8, 67.4, 67.2, 67.1, 63.0, 62.7, 62.1, 61.6, 61.1, 51.5, 49.0, 42.8, 33.8, 29.7, 29.3, 28.8, 28.7, 25.7, 25.6, 25.3, 24.4, 22.4, 21.4, 20.82, 20.75, 20.71, 20.61, 20.56. HR-ESIMS: calcd for C_77_H_89_NO_35_Na: *m*/*z* 1610.5113.; found: *m*/*z* 1610.5152 [M + Na]^+^.

5-(Methoxycarbonyl)pentyl α-d-galactopyranosyl-(1→4)-β-d-galactopyranosyl-(1→3)-β-d-galactopyranosyl-(1→3)-2-acetamido-2-deoxy-α-d-galactopyranoside (**18**)

To a solution of **17** (23 mg, 14.5 mmol) in MeOH (2 mL) was added NaOMe (20 mg) at room temperature and the mixture was stirred at 40 °C for 12 h, then neutralized with Amberlite IR 120 [H^+^]. The mixture was filtered off and concentrated. The product was purified by Sephadex LH-20 column chromatography in MeOH to give **18** (7.3 mg, 60%). ${\left[\alpha \right]}_{D}^{24}$ + 68.2 (*c* = 0.18, MeOH). ^1^H-NMR (500 MHz, CH_3_OH): δ 4.97 (d, 1H, *J*_1, 2_ = 3.7 Hz, H-1 of Gal c), 4.84 (d, 1H, *J*_1, 2_ = 3.7 Hz, H-1 of GalN), 4.55 (d, 1H, *J*_1, 2_ = 7.3 Hz, H-1 of Gal a), 4.49 (d, 1H, *J*_1, 2_ = 7.6 Hz, H-1 of Gal b), 3.66 (s, 3H, OMe), 2.36–2.25 (m, 2H, -CH_2_-), 1.97 (s, 3H, Ac), 1.67–1.60 (m, 4H, -CH_2_-), 1.47–1.38 (m, 2H, -CH_2_-). ^13^C-NMR (125 MHz, CH_3_OH): δ 176.0, 173.9, 106.4 (C-1of Gal b), 105.8 (C-1 of Gal a), 102.6 (C-1 of Gal c), 98.9 (C-1 of GalN), 84.8, 79.6, 79.1, 76.2, 74.4, 73.0, 72.8, 72.1, 71.6, 71.4, 71.1, 70.6, 70.0, 69.6, 68.8, 62.8, 62.7, 62.5, 61.5, 52.0, 50.4, 49.9, 49.5, 49.3, 48.7, 48.5, 34.7, 33.1, 30.8, 30.5, 26.8, 25.8, 23.7, 22.8, 14.4, 1.5. HR-ESIMS: calcd for C_33_H_57_NO_23_Na: *m*/*z* 858.3219.; found: *m*/*z* 858.3225 [M + Na]^+^.

Biotinylated tetrasaccharide (**B**)

Compound **18** (7.3 mg, 8.7 μmol) was dissolved in neat anhydrous ethylenediamine (1.5 mL) and heated at 70 °C for 48 h. The mixture was concentrated with toluene and the product was purified by Sephadex LH-20 column chromatography in H_2_O to give an amine intermediate. The amine was dissolved in DMF (2 mL), and the pH was adjusted to 8–9 using DIPEA. Biotin-NHS (4.0 mg 11.5 μmol) was added and the reaction stirred for 12 h at room temperature. Toluene was added to and evaporated from the residue several times. The product was purified by Sephadex LH-20 column chromatography in H_2_O to give **B** (3.9 mg, 41% 2 steps). ${\left[\alpha \right]}_{D}^{24}$ + 56.9 (*c* = 0.1, H_2_O). ^1^H-NMR (600 MHz, D_2_O): δ 4.84 (br.s, 1H, H-1 of Gal c), 4.77 (br.s, 1H, H-1 of GalN), 4.57 (d, 1H, *J*_1, 2_ = 7.4 Hz, H-1 of Gal a), 4.40 (d, 1H, *J*_1, 2_ = 8.0 Hz, H-1 of Gal b). ^13^C-NMR (150 MHz, D_2_O): δ 215.0, 176.8, 176.6, 174.0, 164.9, 104.0 (C-1 of Gal a,b), 99.9 (C-1 of Gal c), 96.6 (C-1 of GalN), 74.7, 71.7, 70.6, 69.5, 68.5, 68.2, 61.6, 60.1, 59.8, 48.3, 39.3, 38.1, 35.1, 29.8, 27.5, 27.3, 24.7. HRFABMS: calcd for C_44_H_75_N_5_O_24_SNa, *m*/*z* 1112.4420; found, *m*/*z* 1114.4495 [M + Na]^+^.

5-(Methoxycarbonyl)pentyl 2,4,6-tri-*O*-benzyl-3-*O*-chloroacetyl-α-d-galactopyranosyl-(1→3)-2,4,6-tri-*O*-benzoyl-β-d-galactopyranosyl-(1→3)-2-azido-4,6-*O*-benzylidene-2-deoxy-α-d-galactopyranoside (**24**)

Compound **24** was prepared from **8** (100 mg, 0.11 mmol) and **22** (83 mg, 0.13 mmol) as described for preparation of **7**. The product was purified by silica gel column chromatography (10:1 toluene-ethyl acetate) to give **24** (103 mg, 67%). ${\left[\alpha \right]}_{D}^{24}$ +61.8 (*c* = 1.0, CHCl_3_). ^1^H-NMR (500 MHz, CDCl_3_): δ 8.11–7.04 (m, 35H, 7×Ph), 5.89 (d, 1H, *J*_3,4_ = 3.1 Hz, H-4′), 5.76 (br. t, 1H, H-2′), 5.44 (s, 1H, PhCH), 5.28 (d, 1H, *J*_1″, 2″_ = 3.5 Hz, H-1″), 4.93(d, 1H, *J*_1′, 2′_ = 8.0 Hz, H-1′), 4.92 (d, 1H, *J*_1, 2_ = 3.5 Hz, H-1), 3.65 (s, 3H, OMe), 2.35–2.31 (m, 2H, -CH_2_-), 1.69–1.59 (m, 4H, 2×-CH_2_-), 1.42–1.26 (m, 2H, -CH_2_-). ^13^C-NMR (125 MHz, CDCl_3_): δ 174.0, 166.2, 165.98, 165.97, 164.8, 138.2, 138.0, 137.8, 137.7, 133.5, 133.4, 133.0, 130.1, 129.97, 129.72, 129.60, 129.56, 129.1, 128.7, 128.6, 128.5, 128.43, 128.40, 128.3, 128.2, 128.1, 128.13, 127.98 127.86, 127.86, 127.81, 127.6, 127.4, 126.1, 102.9 (C-1′), 100.6, 98.7 (C-1), 93.9 (C-1″), 75.9, 75.3, 75.0, 74.9, 73.9, 73.3, 73.2, 72.7, 72.3, 71.6, 69.0, 68.5, 68.3, 65.9, 63.0, 62.8, 58.4, 51.5, 40.4, 33.9, 29.1, 25.6, 24.6. ESI-HRMS: calcd for C_76_H_78_ClN_3_O_21_Na, 1426.4714 *m*/*z*; found, 1426.4867 *m*/*z* [M + Na]^+^.

5-(Methoxycarbonyl)pentyl 2-*O*-benzoyl-4,6-*O*-benzylidene-3-*O*-chloroacetyl-β-d-galactopyranosyl-(1→3)-2-azido-4,6-*O*-benzylidene-2-deoxy-α-d-galactopyranoside (**25**)

Compound **25** was prepared from **5** (500 mg, 1.18 mmol) and **20** (964 mg, 1.78 mmol) as described for preparation of **7**. The product was purified by silica gel column chromatography (10:1 toluene-ethyl acetate) to give **25** (770 mg, 76%). ${\left[\alpha \right]}_{D}^{24}$ + 56.3 (*c* = 0.5, CHCl_3_). ^1^H-NMR (500 MHz, CDCl_3_): δ 8.02–7.22 (m, 15H, 3×Ph), 5.74 (dd, 1H *J*_2′,3′_ = 10.5, *J*_1′,2′_ = 8.0 Hz, H-2′), 5.55 and 5.54 (each s, 2H, 2×PhCH), 5.22 (dd, 1H, *J*_2,3_ = 10.5 Hz, *J*_3,4_ = 3.5, H-3′), 5.08 (d, 1H, *J*_1′,2′_ = 8.0 Hz, H-1′), 4.94 (d, 1H, *J*_1, 2_ = 3.5 Hz, H-1), 4.50 and 4.49 (each d, 2H, *J*_3,4_ = 3.0Hz, *J*_3′, 4′_ = 3.5Hz, H-4, 4′), 4.40–3.93 (m, 7H, H-3, 6a, 6b, 6′a, 6′b, -CH_2_Cl), 3.81–3.69 (m, 3H, H-5, 5′, -OCH_2_-), 3.66 (s, 3H, OMe), 3.52–3.46 (m, 1H, -OCH_2_-), 2.33–2.30 (m, 2H, -CH_2_-), 1.67–1.61 (m, 4H, 2×-CH_2_-), 1.42–1.36 (m, 2H, -CH_2_-). ^13^C-NMR (125 MHz, CDCl_3_): δ 174.0, 167.3, 165.0, 137.6, 137.4, 133.2, 129.8, 129.5, 129.2, 128.6, 128.4, 128.3, 128.0, 126.4, 126.1, 101.4 (C-1′), 101.1 (PhCH), 100.5 (PhCH), 98.7 (C-1), 75.8, 73.9 (C-3′), 73.6, 73.1, 69.1, 69.0, 68.9, 68.3, 66.3, 58.8, 51.5, 40.7, 33.9, 29.1, 25.7, 24.6. ESI-HRMS: calcd for C_42_H_46_ClN_3_O_14_Na, 874.2566 *m*/*z*; found, 874.2646 *m*/*z* [M + Na]^+^.

5-(Methoxycarbonyl)pentyl 2-*O*-benzoyl-4,6-*O*-benzylidene-β-d-galactopyranosyl-(1→3)-2-azido-4,6-*O*-benzylidene-2-deoxy-α-d-galactopyranoside (**26**)

A solution of **25** (100 mg, 0.12 mmol) in MeOH-Pyr. (3:1, 2 mL) was treated with thiourea (27 mg, 0.35 mmol) under reflux for 2 h. After concentration, the residue was added to the water, extracted with CHCl_3_, and the organic layer was proceeded as usual. The product was purified by silica gel column chromatography (20:1 toluene-acetone) as eluent to give **26** (69 mg, 76%). ${\left[\alpha \right]}_{D}^{24}$ +46.3 (*c* = 0.3, CHCl_3_). ^1^H-NMR (500 MHz, CDCl_3_): δ 8.08–7.23 (m, 15H, 3×Ph), 5.54 and 5.51 (each s, 2H, 2×PhCH), 5.42 (dd, 1H *J*_2′,3′_ = 10.0 Hz, *J*_1′,2′_ = 8.0 Hz, H-2′), 4.96 (d, 1H, *J*_1′,2′_ = 8.0 Hz, H-1′), 4.95 (d, 1H, *J*_1, 2_ = 2.5 Hz, H-1), 4.40–3.93 (m, 6H, H-4, 6a, 6b, 4′, 6′a, 6′b), 3.85 (dt, 1H, H-3), 3.71–3.66 (m, 3H, H-5, 5′, -OCH_2_-), 3.65 (s, 3H, OMe), 3.50–3.44 (m, 1H, -OCH_2_-), 2.73 (d, 1H, OH), 2.34–2.29 (m, 2H, -CH_2_-), 1.69–1.60 (m, 4H, 2×-CH_2_-), 1.42–1.36 (m, 2H, -CH_2_-). ^13^C-NMR (125 MHz, CDCl_3_): δ 174.1, 166.4, 137.7, 137.6, 133.1, 130.0 129.9, 129.3, 128.6, 128.3, 128.3, 128.0, 126.5, 126.1, 101.4, 101.4 (C-1′), 100.6, 98.7 (C-1), 76.0, 75.5, 73.3, 72.9, 71.9, 69.1, 69.0, 68.3, 66.6, 63.1, 58.8, 51.5, 33.9, 29.1, 25.7, 24.7 ESI-HRMS: calcd for C_40_H_45_N_3_O_13_Na, 798.2850 *m*/*z*; found, 798.2912 *m*/*z* [M + Na]^+^.

5-(Methoxycarbonyl)pentyl 2-*O*-benzoyl-4,6-*O*-benzylidene-3-*O*-chloroacetyl-β-d-galactopyranosyl-(1→3)-2-*O*-benzoyl-4,6-*O*-benzylidene-β-d-galactopyranosyl-(1→3)-2-azido-4,6-*O*- benzylidene-2-deoxy-α-d-galactopyranoside (**27**)

Compound **27** was prepared from **26** (300 mg, 0.39 mol) and **20** (417 mg, 0.77 mol) as described for preparation of **7**. The product was purified by silica gel column chromatography (10:1 toluene-ethyl acetate) to give **27** (204 mg, 44%). ${\left[\alpha \right]}_{D}^{24}$ + 81.3 (*c* = 1.0, CHCl_3_). ^1^H-NMR (500 MHz, CDCl_3_): δ 7.86–7.16 (m, 25H, 5×Ph), 5.65–5.58 (m, 2H, H-2 of Gal a,b), 5.48, 5.47 and 5.46 (each s, 3H, 3×PhCH), 5.01 (d, 1H, *J*_1,2_ = 8.0 Hz, H-1 of Gal b), 4.99 (d, 1H, *J*_3, 4_ = 3.0 Hz, H-3 of Gal b), 4.98 (d, 1H, *J*_1, 2_ = 8.0 Hz, H-1 of Gal a), 4.91 (d, 1H, *J*_1, 2_ = 3.5 Hz, H-1 of GalN), 3.66 (s, 3H, OMe), 3.50–3.44 (m, 1H, -OCH_2_-), 2.31–2.28 (m, 2H, -CH_2_-), 1.66–1.55 (m, 4H, 2×-CH_2_-), 1.40–1.35 (m, 2H, -CH_2_-). ^13^C-NMR (125 MHz, CDCl_3_): δ 167.2, 137.6, 137.3, 132.8, 129.7, 129.6, 129.2, 128.6, 128.4, 128.3, 128.2, 128.0, 127.8, 126.3, 126.2, 126.1, 101.4 (C-1 of Gal a), 101.0, 100.7 (C-1 of Gal b), 100.3, 100.1, 98.8 (C-1 of GalN), 75.8, 75.7, 75.4, 73.8, 73.0, 72.3, 69.0, 68.8, 68.2, 66.9, 66.3, 63.2, 58.7, 51.5, 40.6, 29.1, 25.6, 24.6. ESI-HRMS: calcd for C_62_H_64_ClN_3_O_20_Na, 1228.3669 *m*/*z*; found, 1228.3792 *m*/*z* [M + Na]^+^.

5-(Methoxycarbonyl)pentyl 2-*O*-benzoyl-4,6-*O*-benzylidene-β-d-galactopyranosyl-(1→3)-2-*O*-benzoyl-4,6-*O*-benzylidene-β-d-galactopyranosyl-(1→3)-2-azido-4,6-*O*-benzylidene-2-deoxy-α-d-galactopyranoside (**28**)

Compound **28** was prepared from **27** (200 mg, 0.17 mmol) as described for preparation of **11**. The product was purified by silica gel column chromatography (5:1 toluene-ethyl acetate) to give **28** (160 mg, 85%). ${\left[\alpha \right]}_{D}^{24}$ +120.4 (*c* = 0.5, CHCl_3_). ^1^H-NMR (500 MHz, CDCl_3_): δ 7.96–7.17 (m, 25H, 5×Ph), 5.61 (dd, 1H, *J*_1, 2_ = 8.0 Hz, *J*_2, 3_ = 10.0 Hz, H-2 of Gal b), 5.50, 5.48 and 5.42 (each s, 3H, 3×PhCH), 5.31 (dd, 1H, *J*_1, 2a_= 8.1 Hz, *J*_2, 3_ = 9.8 Hz, H-2 of Gal a), 4.98 (d, 1H, *J*_1, 2_ = 8.5 Hz, H-1 of Gal a), 4.91 (d, 1H, *J*_1, 2_ = 8.0 Hz, H-1 of Gal b), 4.91 (d, 1H, *J*_1, 2_ = 4.0 Hz, H-1 of GalN), 3.65 (s, 3H, OMe), 3.50–3.44 (m, 1H, -OCH_2_-), 2.31–2.28 (m, 2H, -CH_2_-), 1.65–1.57 (m, 4H, 2×-CH_2_-), 1.40–1.35 (m, 2H, -CH_2_-). ^13^C-NMR (125 MHz, CDCl_3_): δ 174.1, 166.5, 164.8, 137.7, 137.7, 137.4, 132.9, 132.8, 129.8, 129.7, 129.5, 129.3, 128.7, 128.4, 128.3, 128.2, 127.8, 126.4, 126.3, 126.1, 101.5 (C-1 of Gal a), 101.3, 100.8, 100.3, 99.9 (C-1 of Gal b), 98.8 (C-1 of GalN), 77.3, 75.9, 75.7, 75.5, 75.5, 75.1, 72.5, 72.5, 71.9, 70.9, 69.0, 68.9, 68.8, 68.2, 66.8, 66.7, 63.16, 58.7, 51.5, 33.9, 29.1, 25.6, 24.6. ESI-HRMS: calcd for C_60_H_63_N_3_O_19_Na, 1152.3953 *m*/*z*; found, 1152.4071 *m*/*z* [M + Na]^+^.

5-(Methoxycarbonyl)pentyl 2-*O*-benzoyl-3,6-di-*O*-benzyl-4-*O*-chloroacetyl-β-d-galactopyranosyl-(1→3)-2-*O*-benzoyl-4,6-*O*-benzylidene-β-d-galactopyranosyl-(1→3)-2-*O*-benzoyl-4,6-*O*-benzylidene-β-d-galactopyranosyl-(1→3)-2-azido-4,6-*O*-benzylidene-2-deoxy-α-d-galactopyranoside (**29**)

Compound **29** was prepared from **28** (123 mg, 0.11 mmol) and **9** (83 mg, 0.13 mmol) as described for preparation of **7**. The product was purified by silica gel column chromatography (10:1 toluene-ethyl acetate) to give **29** (94 mg, 53%). ${\left[\alpha \right]}_{D}^{23}$ + 55.3 (*c* = 0.5, CHCl_3_). ^1^H-NMR (500 MHz, CDCl_3_): δ 7.92–6.94 (m, 40H, 8×Ph), 5.57 (d, 1H, *J*_3,4_ = 3.0Hz, H-4 of Gal c), 5.49 (dd, 1H, *J*_1, 2_ = 8.0 Hz, *J*_2, 3_ = 10.0 Hz, H-2 of Gal c), 5.43 (dd, 1H, *J*_1, 2_ = 8.0 Hz, *J*_2, 3_ = 10.0 Hz, H-2 of Gal b), 5.45, 5.32 and 5.28 (each s, 3H, 3×PhCH), 5.26 (dd, 1H, *J*_1, 2_ = 8.5 Hz, *J*_2, 3_ = 9.8 Hz, H-2 of Gal a), 4.92 (d, 1H, *J*_1, 2_ = 7.5 Hz, H-1 of Gal a), 4.90 (d, 1H, *J*_1, 2_ = 3.0 Hz, H-1 of GalN), 4.86 (d, 1H, *J*_1, 2_ = 8.0 Hz, H-1 of Gal b), 4.74 (d, 1H, *J*_1, 2_ = 8.0 Hz, H-1 of Gal c), 3.65 (s, 3H, OMe), 2.32–2.28 (m, 2H, -CH_2_-), 1.65–1.58 (m, 4H, 2×-CH_2_-), 1.39–1.35 (m, 2H, -CH_2_-). ^13^C-NMR (125 MHz, CDCl_3_): δ 174.1, 166.9, 165.0, 164.7, 164.7, 137.8, 137.7, 137.6, 132.8, 132.6, 132.5, 130.3, 129.7, 129.6, 129.5, 129.4, 129.0, 128.8, 128.6, 128.4, 128.4, 128.2, 128.1, 128.1, 128.1, 128.0, 128.0, 127.9, 127.8, 127.8, 127.7, 126.2, 126.2, 126.2, 126.1, 126.1, 101.5 (C-1 of Gal a), 100.6 (C-1 of Gal c), 100.5, 100.4, 99.7 (C-1 of Gal b), 98.8 (C-1 of GalN), 76.2, 75.7, 75.4, 73.7, 73.6, 72.6, 71.7, 71.0, 70.8, 70.6, 69.9, 69.0, 68.8, 68.1, 67.8, 67.0, 66.8, 63.1, 58.6, 51.5, 40.9, 33.9, 29.7, 29.5, 29.0, 25.6, 24.6. ESI-HRMS: calcd for C_89_H_90_ClN_3_O_26_Na, 1675.5477 *m*/*z*; found, 1675.5602 *m*/*z* [M + Na]^+^.

5-(Methoxycarbonyl)pentyl 2-*O*-benzoyl-3,6-di-*O*-benzyl-β-d-galactopyranosyl-(1→3)-2-*O*-benzoyl-4,6-*O*-benzylidene-β-d-galactopyranosyl-(1→3)-2-*O*-benzoyl-4,6-*O*-benzylidene-β-d-galactopyranosyl-(1→3)-2-azido-4,6-*O*-benzylidene-2-deoxy-α-d-galactopyranoside (**30**)

Compound **30** was prepared from **29** (80 mg, 48 mmol) as described for preparation of **11**. The product was purified by silica gel column chromatography (4:1 toluene-EtOAc) to give **30** (74 mg, 97%). ${\left[\alpha \right]}_{D}^{24}$ +64.1 (*c* = 0.5, CHCl_3_). ^1^H-NMR (500 MHz, CDCl_3_): δ 7.86–6.97 (m, 40H, 8×Ph), 5.47–5.32 (m, 3H, H-2 of Gal a, b and c), 5.44, 5.38 and 5.27 (each s, 3H, 3×PhCH), 4.96 (d, 1H, *J*_1, 2_ = 7.5 Hz, H-1 of Gal a), 4.92 (d, 1H, *J*_1, 2_ = 6.5 Hz, H-1 of Gal b), 4.86 (d, 1H, *J*_1, 2_ = 2.5 Hz, H-1 of GalN), 4.65 (d, 1H, *J*_1, 2_ = 7.0 Hz, H-1 of Gal c), 3.64 (s, 3H, OMe), 2.35–2.30 (m, 2H, -CH_2_-), 1.65–1.56 (m, 4H, 2×-CH_2_-), 1.39–1.34 (m, 2H, -CH_2_-). ^13^C-NMR (125 MHz, CDCl_3_): δ 174.1, 165.1, 164.8, 164.7, 137.9, 137.8, 137.8, 137.7, 136.9, 132.7, 132.5, 132.4, 130.1, 129.7, 129.6, 129.5, 129.0, 128.6, 128.5, 128.4, 128.4, 128.3, 128.2, 128.1, 128.0, 128.0, 127.9, 127.9, 127.8, 127.8, 127.6, 127.5, 126.3, 126.3, 126.2, 125.3, 101.6(C-1 of Gal a), 100.9 (C-1 of Gal c), 100.6, 100.5, 100.4, 99.8 (C-1 of Gal b), 98.8 (C-1 of GalN), 75.7, 75.5, 75.3, 73.6, 73.2, 71.2, 70.8, 70.7, 70.2, 69.0, 68.9, 68.7, 66.7, 66.7, 65.8, 63.1, 58.5, 51.5, 33.9, 29.7, 29.0, 25.6, 24.6, 21.4. ESI-HRMS: calcd for C_87_H_89_N_3_O_25_Na, 1598.5683 *m*/*z*; found, 1598.5839 *m*/*z* [M + Na]^+^.

5-(Methoxycarbonyl)pentyl 2,3-di-*O*-benzyl-4,6-*O*-di-*tert*-butylsilylene-α-d-galactopyranosyl-(1→4)-2-*O*-benzoyl-3,6-di-*O*-benzyl-β-d-galactopyranosyl-(1→3)-2-*O*-benzoyl-4,6-*O*-benzylidene-β-d-galactopyranosyl-(1→3)-2-*O*-benzoyl-4,6-*O*-benzylidene-β-d-galactopyranosyl-(1→3)-2-azido-4,6-*O*-benzylidene-2-deoxy-α-d-galactopyranoside (**31**)

Compound **31** was prepared from **30** (64 mg, 41 mmol) and **12** (36 mg, 61 mmol) as described for preparation of **7**. The product was purified by silica gel column chromatography (5:1 toluene-ethyl acetate) to give **31** (57 mg, 68%). ${\left[\alpha \right]}_{D}^{24}$
**+** 124.5 (*c* = 1.0, CHCl_3_). ^1^H-NMR (500 MHz, CDCl_3_): δ 7.94–6.96 (m, 50H, 10×Ph), 5.50–5.40 (m, 3H, H-2 of Gal a, b and c), 5.45, 5.32 and 5.25 (each s, 3H, 3×PhCH), 4.94 (d, 1H, *J*_1, 2_ = 8.0 Hz, H-1 of Gal a), 4.92 (d, 1H, *J*_1, 2_ = 3.5 Hz, H-1 of GalN), 4.92 (d, 1H, *J*_1, 2_ = 7.5 Hz, H-1 of Gal b), 4.74 (d, 1H, *J*_1, 2_ = 4.0 Hz, H-1 of Gal d), 4.63 (d, 1H, *J*_1, 2_ = 7.5 Hz, H-1 of Gal c), 3.65 (s, 3H, OMe), 2.35–2.31 (m, 2H, -CH_2_-), 1.67–1.56 (m, 4H, 2×-CH_2_-), 1.37–1.34 (m, 2H, -CH_2_-), 0.96 and 0.92 (each s, 18H, 2×(CH_3_)_3_). ^13^C-NMR (125 MHz, CDCl_3_): δ 174.1, 165.1, 164.8, 164.7, 137.9, 137.85, 137.80, 137.7, 132.7, 132.5, 132.5, 130.2, 129.68, 129.62, 129.5, 129.0, 128.6, 128.5, 128.41, 128.36, 128.29, 128.23. 128.1, 128.02, 127.97, 127.93, 127.90, 127.84, 127.78, 127.6, 127.5, 126.31, 126.2, 125.3, 101.6 (C-1 of Gal a), 100.87, 100.6 (C-1 of Gal c), 100.5, 100.3 (C-1 of Gal b), 99.8 (C-1 of Gal d), 98.8 (C-1 of GalN), 78.7, 77.8, 75.9, 75.7, 75.5, 75.2, 74.4, 73.6, 73.6, 73.4, 73.2, 72.3, 71.2, 70.8, 70.6, 70.5, 70.1, 70.0, 69.0, 68.8, 68.5, 68.2, 67.6, 67.5, 67.1, 66.8, 63.1, 58.6, 51.5, 33.9, 29.0, 27.6, 27.4, 27.3, 27.3, 25.6, 24.6, 23.3, 21.5, 20.7. ESI-HRMS: calcd for C_115_H_127_N_3_O_30_SiNa, 2080.8171 *m*/*z*; found, 2080.8264 *m*/*z* [M + Na]^+^.

2-(Trimethylsilyl)ethyl 2,3,4,6-tetra-*O*-benzyl-α-d-galactopyranosyl-(1→4)-2-*O*-benzoyl-3,6-di-*O*-benzyl-β-d-galactopyranoside (**34**)

To a solution of **32** (145 mg, 0.26 mmol) and **33** (195 mg, 0.31 mmol) in dry CH_2_Cl_2_—toluene (1:1, 1.6 mL) powdered AW300 (330 mg) was added, and the mixture was stirred under Ar atmosphere at room temperature for 2 h, then cooled to –10 °C. NIS (139 mg, 0.62 mmol) and TfOH (2.7 μL, 31 μmol) were added to the mixture, which was stirred at −10 °C for 10min., then neutralized with Et_3_N. The precipitates were filtered off and washed with CHCl_3_. The combined filtrate and washings were successively washed with saturated aqueous Na_2_S_2_O_3_ and water, dried (MgSO_4_), and concentrated. The product was purified by silica gel column chromatography (5:1 n-hexane-EtOAc) to give **34** (222 mg, 79%). ${\left[\alpha \right]}_{D}^{24}$ +57.6 (*c* 1.2, CHCl_3_). ^1^H-NMR (500 MHz, CDCl_3_): δ 8.03–7.05 (35H, m, 7×Ph), 5.52 (1H, dd, *J*_1, 2_ = 8.0 Hz, *J*_2, 3_ = 7.9 Hz, H-2), 5.03 (1H, d, *J*_1′, 2′_ = 3.4 Hz, H-1′), 4.96–4.89 (2H, m, 2×PhCH_2_), 4.80 (2H, s, PhCH_2_), 4.72–4.66 (2H, m, 2×PhCH_2_),4.58–4.53 (2H, m, H-5, PhCH_2_), 4.48 (1H, d, H-1), 4.34 (1H, d, *J*_gem_ = 12.9 Hz, PhCH_2_), 4.27–4.12 (8H, m, H-4, H-2′, H-3′, H-4′, 4×PhCH_2_), 4.06 (1H, dd, *J*_5, 6a_ = 1.2 Hz, *J*_6a, 6b_ = 10.9 Hz, H-6a), 4.04–3.94 (1H, m, CH_2_), 3.59–3.49 (5H, m, H-3, H-6a’, H-6b’, H-5′, CH_2_), 3.22 (1H, dd, *J*_5, 6b_ = 3.9 Hz, H-6b), 0.91–0.81 (2H, m, 2×CH_2_), −0.10 (9H, s, TMS). ^13^C-NMR (125 MHz, CDCl_3_): δ 165.1, 139.0, 138.9, 138.7, 138.4, 138.0, 137.8, 132.8, 130.4, 129.8 × 2, 128.3 × 2, 128.22 × 2, 128.17 × 2, 128.12 × 2, 128.09 × 2, 128.06 × 2, 128.04 × 2, 128.03 × 2, 127.9 × 2, 127.67, 127.64 × 2, 127.63 × 2, 127.60, 127.5, 127.4 × 2, 127.33, 127.30, 127.26, 127.22 × 2, 101.03 (C-1), 100.96 (C-1′), 79.3 (PhCH_2_), 78.6 (C-3), 76.4 (C-3′), 74.9 (PhCH_2_), 74.7 (C-4), 73.9 (PhCH_2_), 73.8 (C-5′), 73.7 (PhCH_2_), 73.0 (C-4′), 72.8 (C-2′), 72.3 (PhCH_2_), 71.3 (C-2), 71.0 (PhCH_2_), 68.9 (C-5), 67.7 × 2 (C-6, C-6′), 66.9 (CH_2_), 17.9 (CH_2_), −1.5 × 3 (TMS). ESI-HRMS: calcd for C_66_H_74_O_12_SiNa, 1109.4847 *m*/*z*; found, 1109.4930 *m*/*z* [M + Na]^+^.

2-(Trimethylsilyl)ethyl 2,3,4,6-tetra-*O*-acetyl-α-d-galactopyranosyl-(1→4)-2-*O*-benzoyl-3,6-di-*O*-acetyl-β-d-galactopyranoside (**35**)

To a solution of **34** (2.47 g, 2.27 mmol) in THF—MeOH (1:1, 10.0 mL) was hydrogenolysed under hydrogen in the presence of Pd(OH)_2_/C (1.05 g) for 20 h at room temperature. The mixture was filtered and concentrated, and the residue was acetylated with acetic anhydride (23 mL) in pyridine (15 mL) at 50 °C for 4 h. After the reaction was quenched with MeOH (20 mL) at 0 °C, toluene was added and co-evaporated several times. The product was purified by silica gel column chromatography (2:1 n-hexane-EtOAc) to give **35** (1.55 g, 85%). ${\left[\alpha \right]}_{D}^{24}$ +73.4 (*c* 1.2, CHCl_3_). ^1^H-NMR (500 MHz, CDCl_3_): δ 7.99–7.18 (5H, m, Ph), 5.61 (1H, d, *J*_3′, 4′_ = 2.4 Hz, H-4′), 5.47–5.42 (2H, m, H-2, H-3′), 5.22 (1H, dd, *J*_1′, 2′_ = 3.5 Hz, H-2′), 5.05 (1H, d, H-1′), 5.02 (1H, dd, *J*_2, 3_ = 10.8 Hz, H-3), 4.65 (1H, d, *J*_1, 2_ = 7.7 Hz, H-1), 4.64–4.62 (1H, m, H-5), 4.51 (1H, dd, *J*_5′, 6a’_ = 6.6 Hz, *J*_6a’, 6b’_ = 11.1 Hz, H-6a’), 4.21–4.14 (3H, m, H-6a, H-6b, H-6b’), 4.11 (1H, d, *J*_3, 4_ = 2.3 Hz, H-4), 4.03–3.97 (1H, m, CH_2_), 3.86 (1H, t, *J*_5′, 6b’_ = 6.6 Hz, H-5′), 3.60–3.55 (1H, m, CH_2_), 2.14, 2.12, 2.09, 2.05, 2.00 and 1.96 (18H, each s, 6×Ac), 0.96–0.86 (2H, m, CH_2_), −0.07 (9H, s, TMS). ^13^C-NMR (125 MHz, CDCl_3_): δ 170.7, 170.6, 170.5 × 2, 170.1, 169.7, 164.8, 133.1, 129.7 × 2, 129.5, 128.4 × 2, 100.7 (C-1), 99.2 (C-1′), 77.0 (C-4), 72.7 (C-3), 71.8 (C-5′), 69.3 (C-2), 68.6 (C-2′), 67.9 (C-4′), 67.42 (CH_2_), 67.36 (C-3′), 67.1 (C-5), 61.9, (C-6′), 60.7 (C-6), 20.8, 20.74, 20.69, 20.68, 20.6 × 2, 17.8 (CH_2_), −1.5 × 3 (TMS). ESI-HRMS: calcd for C_36_H_50_O_18_SiNa, 821.2664 *m*/*z*; found, 821.2721 *m*/*z* [M + Na]^+^.

2,3,4,6-Tetra-*O*-acetyl-α-d-galactopyranosyl-(1→4)-2-*O*-benzoyl-3,6-di-*O*-acetyl-α-d-galactopyranosyl trichloroacetimidate (**36**)

To a solution of **35** (0.92 g, 1.15 mmol) in CH_2_Cl_2_ (8.5 mL), cooled to 0 °C was added CF_3_CO_2_H (8.5 mL), and the mixture was stirred at room temperature for 0.5 h and concentrated. EtOAc and toluene (1:2) were added and evaporated to give the reducing sugar. To a solution of the residue in CH_2_Cl_2_ (10.0 mL) cooled at 0 °C DBU (160 μL, 1.05 mmol) and CCl_3_CN (1.3 mL, 1.29 mmol) were added. The reaction mixture was stirred at room temperature for 0.5 h. After completion of the reaction, the mixture was concentrated. The residue was purified by silica gel column chromatography (3:2 n-hexane-EtOAc) to give **36** (0.81 g, 84%). ${\left[\alpha \right]}_{D}^{24}$ +110.0 (*c* 1.2, CHCl_3_). ^1^H-NMR (500 MHz, CDCl_3_): δ 8.61(1H, s, NH), 7.97—7.47 (5H, m, Ph), 6.74 (1H, d, *J*_1, 2_ = 3.6 Hz, H-1), 5.68 (1H, dd, *J*_2, 3_ = 11.0 Hz, H-2), 5.60 (1H, d, *J*_3′, 4′_ = 2.2 Hz, H-4′), 5.57 (1H, dd, *J*_3, 4_ = 2.5 Hz, H-3), 5.42 (1H, dd, *J*_2′, 3′_ = 11.0 Hz, H-3′), 5.27 (1H, dd, *J*_1′, 2′_ = 3.7 Hz H-2′), 5.08 (1H, d, H-1′), 4.63 (1H, t, *J*_5, 6a_ = *J*_5, 6b_ = 6.8 Hz, H-5), 4.42–4.35 (2H, m, H-5′, H-6a’), 4.32 (1H, d, H-4), 4.18–4.09 (3H, m, H-6a, H-6b, H-6b’), 2.15, 2.14, 2.05, 2.03, 2.01 and 2.00 (each s, 18H, 6×Ac). ^13^C-NMR (125 MHz, CDCl_3_): δ 170.68, 170.63, 170.4×2, 170.3, 170.0, 165.4, 160.6, 133.7, 129.9×2, 129.0, 128.7×2, 99.0 (C-1′), 93.8 (C-1), 90.9, 76.7 (C-4), 70.9 (C-5′), 69.4 (C-3′), 68.3 (C-2′), 68.0 (C-3), 67.5×2 (C-2, C-5), 67.3 (C-4′), 61.9 (C-6′), 61.0 (C-6), 21.0, 20.9, 20.85, 20.78×2, 20.7. ESI-HRMS: calcd for C_33_H_38_Cl_3_NO_18_Na, 864.1052 *m*/*z*; found, 864.1087 *m*/*z* [M + Na]^+^.

5-(Methoxycarbonyl)pentyl 2,3,4,6-tetra-*O*-acetyl-α-d-galactopyranosyl-(1→4)-2-*O*-benzoyl-3,6-di-*O*-acetyl-β-d-galactopyranosyl-(1→3)-2-*O*-benzoyl-4,6-*O*-benzylidene-β-d-galactopyranosyl-(1→3)-2-azido-4,6-*O*-benzylidene-2-deoxy-α-d-galactopyranoside (**37**)

A solution of **26** (154 mg, 0.20 mmol) and **36** (211 mg, 0.25 mmol) containing activated MS-AW300 (550 mg) in dry CH_2_Cl_2_ (1.5 mL) was stirred under an atmosphere of argon at room temperature for 18 h, then cooled to −40 °C. TMSOTf (2.5 μL, 13.8 μmol) was added, and the mixture was stirred for 3 h at room temperature, then **36** (74 mg, 0.088 mmol) and TMSOTf (2.5 μL, 14 μmol) were added, and the mixture was stirred at −20 °C for 3 h. After the reaction, they were neutralized with Et_3_N. The solids were filtrated off and washed with CHCl_3_. The combined filtrate and washings were successively washed with brine, dried (MgSO_4_), and concentrated. The product was purified by flash silica gel column chromatography using 4:1 toluene-acetone as eluent to give **37** (99 mg, 34%). ${\left[\alpha \right]}_{D}^{24}$ +109.0 (*c* 1.1, CHCl_3_). ^1^H-NMR (500 MHz, CDCl_3_): δ 8.08–7.19 (20H, m, 4×Ph), δ 5.63–7.19 (20H, m, 4×Ph), 5.51 (1H, s, PhCH), 5.49 (1H, s, PhCH), 5.02 (1H, d, *J*_1, 2_ = 3.3 Hz, H-1 of Gal c), 5.01 (1H, d, *J*_1, 2_ = 8.1 Hz, H-1 of Gal b), 4.96 (1H, d, *J*_1, 2_ = 7.5 Hz, H-1 of Gal a), 4.92 (1H, d, *J*_1, 2_ = 3.5 Hz, H-1 of GalN), 3.67 (3H, s, OMe), 2.29 (2H, t, -CH_2_-), 2.13, 2.07, 2.03, 2.01, 2.00 and 1.86 (18H, 6×Ac), 1.65–1.59 (4H, m, 2×-CH_2_-), 1.40–1.26 (2H, m, -CH_2_-). ^13^C-NMR (125 MHz, CDCl_3_): δ 174.0, 170.50, 170.47, 170.4, 170.2, 170.1, 170.0, 164.8, 164.6, 137.63, 137.55, 133.0, 132.7, 129.8, 129.53 × 2, 129.52 × 2, 129.0, 128.7, 128.3, 128.2 × 2, 128.1 × 2, 128.0 × 2, 127.8 × 2, 126.2 × 2, 126.0 × 2, 101.3 (C-1 of Gal a), 100.8 (PhCH), 100.6 (C-1 of Gal b), 100.3 (PhCH), 99.9 (C-1 og Gal c), 98.8 (C-1 of GalN), 76.1, 76.0, 75.7, 75.6, 72.5, 72.23, 72.17, 70.7, 69.0 × 2, 68.9, 68.5, 68.2, 67.9, 67.2, 67.1, 66.7, 63.1, 61.6, 60.8, 58.6, 61.5, 33.8, 29.0, 25.6, 24.6, 20.8, 20.73, 20.70, 20.66, 20.63, 20.61. ESI-HRMS: calcd for C_71_H_81_N_3_O_30_Na, 1478.4803 *m*/*z*; found, 1478.4878 *m*/*z* [M + Na]^+^.

5-(Methoxycarbonyl)pentyl 2,3,4,6-tetra-*O*-acetyl-α-d-galactopyranosyl-(1→4)-2-*O*-benzoyl-3,6-di-*O*-acetyl-β-d-galactopyranosyl-(1→3)-2-*O*-benzoyl-4,6-di*-O*-acetyl-β-d-galactopyranosyl-(1→3)-2-azido-4,6-di-*O*-acetyl-2-deoxy-α-d-galactopyranoside (**38**)

A solution of **37** (106 mg, 72.8 μmol) in 80% AcOH (5.0 mL) was stirred at 70 °C for 6 h. Toluene was added and co-evaporated several times. The residue was acetylated with acetic anhydride (1.0 mL) in pyridine (1.0 mL). After the reaction was quenched with MeOH (20 mL) at 0 °C, the reaction mixture was added toluene and concentrated. The residue was purified by silica gel column chromatography using 1:1 CHCl_3_—EtOAc as eluent to give **38** (78 mg, 74%). ${\left[\alpha \right]}_{D}^{24}$ +104.5 (*c* 1.3, CHCl_3_). ^1^H-NMR (500 MHz, CDCl_3_): δ 7.67–7.16 (10H, m, 2×Ph), 5.60 (1H, d, *J*_3, 4_ = 2.5 Hz, H-4 of Galc), 5.53–5.50 (2H, m, H-4 of Gala, H-3of Gal c), 5.44 (1H, d, *J*_3, 4_ = 2.9 Hz, H-4 of GalN), 5.35–5.31 (2H, m, H-2 of Gal a,b), 5.01 (1H, d, *J*_1, 2_ = 3.7 Hz, H-1 of Gal c), 4.82 (1H, d, *J*_1, 2_ = 3.7 Hz, H-1 of GalN), 4.77 (1H, d, *J*_1, 2_ = 7.8 Hz, H-1 of Gal a), 4.74 (1H, d, *J*_1, 2_ = 3.5 Hz, H-1 of Gal b). 3.66 (3H, s, OMe), 2.31 (2H, t,-CH_2_-), 2.36, 2.19, 2.15, 2.13, 2.11, 2.09. 2.04, 2.02, 1.99 and 1.84 (30H, 10×Ac), 1.64–1.58 (m, 4H, 2×-CH_2_-), 1.39–1.26 (m, 2H, -CH_2_-). ^13^C-NMR (125 MHz, CDCl_3_): δ 173.9, 170.8, 170.7, 170.6, 170.5 × 3, 170.1, 169.9, 169.7, 169.4, 164.5, 164.4, 132.8, 129.41 × 2, 129.38 × 2, 129.3, 129.02, 128.21 × 2, 128.16 × 2, 128.1, 101.5 (C-1 of Gal a), 101.2 (C-1 of Gal b), 99.0 (C-1 of Gal c), 97.9 (C-1 of GalN), 76.6, 76.3, 73.4, 72.1, 72.0, 71.5, 70.9, 69.5, 69.2, 69.1, 68.8, 68.2, 68.1, 67.4, 67.2, 67.1, 62.8, 62.3, 61.6, 60.7, 59.2, 51.5, 33.8, 28.9, 25.6, 24.5, 21.5, 20.82, 20.75, 20.73 × 2, 20.69 × 2, 20.66 × 2, 20.6. ESI-HRMS: calcd for C_65_H_81_N_3_O_34_Na, 1470.4599 *m*/*z*; found, 1470.4568 *m*/*z* [M + Na]^+^.

5-(Methoxycarbonyl)pentyl 2,3,4,6-tetra-*O*-acetyl-α-d-galactopyranosyl-(1→4)-2-*O*-benzoyl-3,6-di-*O*-acetyl-β-d-galactopyranosyl-(1→3)-2-*O*-benzoyl-4,6-di*-O*-acetyl-β-d-galactopyranosyl-(1→3)-2-acetamido-4,6-di-*O*-acetyl-2-deoxy-α-d-galactopyranoside (**39**)

To a solution of **38** (77 mg, 53 μmol) in THF—H_2_O (6:1, 3.5 mL) triphenylphosphine (PPh_3_) (15.8 mg, 60 μmol) was added. The mixture was stirred at 70 °C for 6 h. After completion of the reaction, the mixture was diluted with EtOAc, washed with saturated aqueous NaHCO_3_ and water, dried (MgSO_4_), and concentrated. The residue was acetylated with acetic anhydride (2.0 mL) in pyridine (3.0 mL). After the reaction was quenched with MeOH, toluene was added and co-evaporated several times. The product was purified by silica gel column chromatography (2:1 toluene-acetone) to give **39** (65 mg, 83%). ${\left[\alpha \right]}_{D}^{24}$ +102.8 (*c* 1.0, CHCl_3_). ^1^H-NMR (500 MHz, CDCl_3_): δ 7.66–7.15 (10H, m, 2×Ph), 5.60 (1H, d, *J*_3, 4_ = 2.5 Hz, H-4 of Galc), 5.54–5.26 (7H, m, H-4 of GalN, H-2,4 of Gala, H-2,3 of Galb, H-3,4 of Galc), 5.18 (1H, dd, *J*_1, 2_ = 3.5 *J*_2, 3_ = 11.0Hz, H-2 of GalN), 5.01 (1H, d, *J*_1, 2_ = 3.9 Hz, H-1 of Gal c), 4.88 (1H, d, *J*_1, 2_ = 3.2 Hz, H-1 of GalN), 4.73 (1H, d, *J*_1, 2_ = 7.3 Hz, H-1 of Gal a), 4.68 (1H, d, *J*_1, 2_ = 7.8 Hz, H-1 of Gal b), 3.68 (3H, s, OMe), 2.26 (2H, t, -CH_2_-), 2.19, 2.14, 2.13, 2.11, 2.10, 2.03, 2.01, 1.99, 1.83, 1.66 and 1.54 (33H, 10×Ac), 1.57–1.53 (4H, m, 2×-CH_2_-), 1.46–1.34 (2H, m, -CH_2_-). ^13^C-NMR (125 MHz, CDCl_3_): δ 174.0, 170.8, 170.7, 170.58, 170.55, 170.5, 170.4, 170.10, 170.07, 170.0, 169.9, 169.5, 164.6, 164.4, 132.3, 132.8, 129.42 × 2, 129.37 × 2, 129.0, 128.9, 128.4 × 2, 128.2 × 2, 101.2 (C-1 of Gal a), 99.7 (C-1 of Gal b), 99.1 (C-1 of Gal c), 97.0 (C-1 of GalN), 76.3, 74.1, 72.1, 71.9, 71.8, 69.2, 69.0, 68.1, 67.8, 67.6, 67.2, 67.12, 67.14, 67.12, 63.0, 62.5, 61.5, 60.7 × 2, 51.5 × 2, 49.1, 33.8, 28.7, 25.6, 24.5, 22.5, 20.82, 20.79 × 2, 20.76, 20.73 × 2, 20.68 × 2, 20.6 × 2. ESI-HRMS: calcd for C_67_H_85_NO_35_Na, 1486.4780 *m*/*z*; found, 1486.4758 *m*/*z* [M + Na]^+^.

5-(Methoxycarbonyl)pentyl α-d-galactopyranosyl-(1→4)-β-d-galactopyranosyl-(1→3)-β-d-galactopyranosyl-(1→3)-2-acetamido-2-deoxy-α-d-galactopyranoside (**18**)

To a solution of **39** (62 mg, 42.3 μmol) in MeOH (1.0 mL) 1,4-dioxane (1.0 mL) and NaOMe (18 mg) was added at 45 °C. The mixture was stirred for 17 h and then neutralized with Amberlite IR 120 [H^+^]. The mixture was filtered and concentrated. The product was purified by Sephadex LH-20 column chromatography in MeOH to give **18** (34.5 mg, 98%). Spectral data is described in the experimental part for synthesizing **17** to **18**.

Biotinylated tetrasaccharide (**B**)

Compound **18** (23.6 mg, 28.2 μmol) was dissolved in neat anhydrous ethylenediamine (4.8 mL) and heated at 70 °C for 64 h. The mixture was concentrated with toluene and the product was purified by Sephadex LH-20 column chromatography in H_2_O to give an amine intermediate. The amine derivative was dissolved in DMF (6 mL), and the pH was adjusted to 8–9 using DIPEA. Biotin-NHS (14.4 mg, 41.4 μmol) was added and the reaction was stirred at room temperature for 19 h. Toluene was added to and evaporated from the residue several times. The product was purified by Sephadex LH-20 column chromatography in H_2_O to give **B** (29.0 mg, 94%). Spectral data is described above **B**.

5-(Methoxycarbonyl)pentyl 2,3,4,6-tetra-*O*-acetyl-α-d-galactopyranosyl-(1→4)-2-*O*-benzoyl-3,6-di-*O*-acetyl-β-d-galactopyranosyl-(1→3)-2-*O*-benzoyl-4,6-*O*-benzylidene-β-d-galactopyranosyl-(1→3)-2-*O*-benzoyl-4,6-*O*-benzylidene-β-d-galactopyranosyl-(1→3)-2-azido-4,6-*O*-benzylidene-2-deoxy-α-d-galactopyranoside (**40**)

Compound **40** was prepared from **28** (436 mg, 0.39 mmol) and **36** (813 mg, 0.96 mmol) as described for preparation of **37**. The product was purified by silica gel column chromatography (1:1 CHCl_3_—EtOAc) to give **40** (342 mg, 49%). ${\left[\alpha \right]}_{D}^{23}$ +107.1 (*c* 1.2, CHCl_3_). ^1^H-NMR (500 MHz, CDCl_3_): δ 7.96–7.12 (30H, m, 6×Ph), 5.47. 5.46 and 5.31 (3H, each s, 3×PhCH), 4.97 (1H, d, *J*_1, 2_ = 3.7 Hz, H-1 of Gal d), 4.94 (1H, d, *J*_1, 2_ = 7.3 Hz, H-1 of Gal c), 4.93 (1H, d, *J*_1, 2_ = 7.3 Hz, H-1 of Gal a), 4.90 (1H, d, *J*_1, 2_ = 3.3 Hz, H-1 of GalN), 4.83 (1H, d, *J*_1, 2_ = 7.5 Hz, H-1 of Gal b), 3.65 (3H, s, OMe), 2.29 (2H, t, -CH_2_-), 2.11, 2.03, 1.99, 1.98, 1.97 and 1.82 (18H, 6×Ac), 1.63–1.60 (4H, m, 2×-CH_2_-), 1.41–1.26 (2H, m, -CH_2_-). ^13^C-NMR (125 MHz, CDCl_3_): δ 174.1, 170.51, 170.47, 170.4, 170.2, 170.1, 169.7, 164.8×2, 164.7, 137.74, 137.70, 137.6, 133.0, 132.7, 132.6, 130.3, 129.7 × 2, 129.6, 129.55 × 2, 129.50 × 2, 129.4, 129.0, 128.8 × 2, 128.43, 128.36 × 2, 128.27 × 2, 128.12 × 2, 128.12 × 2, 128.05 × 2, 127.8 × 2, 126.3 × 2, 126.2 × 2, 126.1 × 2, 101.5 (C-1 of Gal a), 100.8 (C-1 of Gal b), 100.7 (PhCH), 100.6 (PhCH), 100.4 (PhCH), 99.6 (C-1 of Gal c), 98.9 (C-1 of Gal d), 98.8 (C-1 of GalN), 76.1, 75.74, 75.70, 75.4, 73.8, 72.5, 72.4, 72.2, 70.8, 70.3, 69.1 × 2, 68.8, 68.6, 68.5, 68.2, 67.9, 67.2, 67.1, 66.81, 66.77, 63.1, 61.5, 60.8, 58.6, 51.5, 33.9, 29.7, 29.0, 25.6, 24.6, 20.8, 20.68, 20.65 × 2, 20.6. ESI-HRMS: calcd for C_91_H_99_N_3_O_36_K, 1848.5645 *m*/*z*; found, 1848.5612 *m*/*z* [M + K]^+^.

5-(Methoxycarbonyl)pentyl 2,3,4,6-tetra-*O*-acetyl-α-d-galactopyranosyl-(1→4)-2-*O*-benzoyl-3,6-di-*O*-acetyl-β-d-galactopyranosyl-(1→3)-2-*O*-benzoyl-4,6-di*-O*-acetyl-β-d-galactopyranosyl-(1→3)-2-*O*-benzoyl-4,6-di*-O*-acetyl-β-d-galactopyranosyl-(1→3)-2-azido-4,6-di-*O*-acetyl-2-deoxy-α-d-galactopyranoside (**41**)

Compound **41** was prepared from **40** (363 mg, 0.20 mmol) by the same method described for preparation of **38**. The product was purified by silica gel column chromatography (1:1 CHCl_3_—EtOAc) to give **41** (295 mg, 82%). ${\left[\alpha \right]}_{D}^{23}$ +91.6 (*c* 1.0, CHCl_3_). ^1^H-NMR (500 MHz, CDCl_3_): δ 7.71–7.04 (15H, m, 3×Ph), 4.98 (1H, d, *J*_1, 2_ = 3.4 Hz, H-1 of Gal d), 4.80 (1H, d, *J*_1, 2_ = 3.7 Hz, H-1 of GalN), 4.69 (1H, d, *J*_1, 2_ = 7.8 Hz, H-1 of Gal a), 4.63 (1H, d, *J*_1, 2_ = 7.8 Hz, H-1 of Gal c), 4.58 (1H, d, *J*_1, 2_ = 7.6 Hz, H-1 of Gal b), 3.65 (3H, s, OMe), 2.31 (2H, t, -CH_2_-), 2.17, 2.12×2, 2.07, 2.063, 2.056, 2.045. 2.02, 2.01, 2.00, 1.98 and 1.80 (36H, 12×Ac), 1.64–1.56 (4H, m, 2×-CH_2_-), 1.36–1.25 (2H, m, -CH_2_-). ^13^C-NMR (125 MHz, CDCl_3_): δ 174.0, 170.8, 170.79, 170.74, 170.66, 170.6 × 2, 170.5, 170.2, 170.1, 170.0, 169.8, 169.5, 164.5, 164.3, 164.1, 133.0, 132.8, 132.6, 129.6 × 3, 129.4 × 2, 129.3 × 2, 129.1, 129.0, 128.3 × 2, 128.2 × 2, 128.1 × 2, 101.7 (C-1 of Gal a), 101.1 (C-1 of Gal b), 101.0 (C-1 of Gal c), 99.1 (C-1 of Gal d), 98.0 (C-1 of GalN), 76.3 × 2, 75.4, 73.5, 72.1, 71.8, 71.5, 71.4, 71.0, 69.6, 69.2 × 2, 69.1, 68.9, 68.3, 68.2, 67.4, 67.3, 67.2, 62.9, 62.4, 62.1, 61.6, 60.8, 60.5, 59.3, 51.6, 33.9, 29.0, 25.7, 24.6, 20.89 × 2, 20.87, 20.80 × 2, 20.78 × 2, 20.75 × 3, 20.69, 20.68. ESI-HRMS: calcd for C_82_H_99_N_3_O_42_Na, 1820.5601 *m*/*z*; found, 1820.5710 *m*/*z* [M + Na]^+^.

5-(Methoxycarbonyl)pentyl 2,3,4,6-tetra-*O*-acetyl-α-d-galactopyranosyl-(1→4)-2-*O*-benzoyl-3,6-di-*O*-acetyl-β-d-galactopyranosyl-(1→3)-2-*O*-benzoyl-4,6-di*-O*-acetyl-β-d-galacto-pyranosyl-(1→3)-2-*O*-benzoyl-4,6-di*-O*-acetyl-β-d-galactopyranosyl-(1→3)-2-acetamido-4,6-di-*O*-acetyl-2-deoxy-α-d-galactopyranoside (**42**)

Compound **42** was prepared from **41** (279 mg, 0.16 mmol) by the same method described for preparation of **39**. The product was purified by silica gel column chromatography (3:2 toluene-acetone) to give **42** (232 mg, 82%). [α]_D_ +98.0 (*c* 1.2, CHCl_3_). ^1^H-NMR (500 MHz, CDCl_3_): δ 7.69–7.03 (15H, m, 3×Ph), 4.98 (1H, d, *J*_1, 2_ = 3.5 Hz, H-1 of Gal d), 4.85 (1H, d, *J*_1, 2_ = 3.7 Hz, H-1 of GalN), 4.62 (1H, d, *J*_1, 2_ = 8.0 Hz, H-1 of Gal c), 4.58 (2H, m, H-1 of Gal a, H-1 of Gal b), 3.67 (3H, s, OMe), 2.35 (2H, t, -CH_2_-), 2.17, 2.12×2, 2.08, 2.07, 2.06, 2.01. 2.004, 1.997, 1.98, 1.80 and 1.51 (36H, each s, 12×Ac), 1.58–1.52 (4H, m, 2×-CH_2_-), 1.38–1.20 (2H, m, -CH_2_-). ^13^C-NMR (125 MHz, CDCl_3_): δ 174.0, 170.7 × 2, 170.5 × 2, 170.4 × 2, 170.1, 170.0 × 2, 169.9 × 2, 169.8, 169.4, 164.4 × 2, 163.9, 133.3, 132.7, 132.5, 129.5 × 2, 129.3 × 2, 129.13 × 2, 129.1, 128.8, 128.5 × 2, 128.2, 128.1 × 2, 128.0 × 2, 100.1 (C-1 of Gal a), 100.8 (C-1 of Gal b), 99.8 (C-1 of Gal c), 99.0 (C-1 of Gal d), 96.9 (C-1 of GalN), 76.1 × 2, 75.6, 74.1, 72.2, 72.0, 71.7, 71.6, 71.4, 70.9, 69.1, 69.0, 68.8 × 2, 68.1, 67.7, 67.6, 67.2, 67.1 × 2, 63.0, 62.2, 62.1, 61.4, 60.6, 51.5, 49.1, 33.7, 28.7, 25.5, 24.4, 22.5, 20.8 × 3, 20.71, 20.67, 20.63 × 2, 20.61 × 3, 20.56 × 2. ESI-HRMS: calcd for C_84_H_103_NO_43_Na, 1836.5802 *m*/*z*; found, 1836.5949 *m*/*z* [M + Na]^+^.

5-(Methoxycarbonyl)pentyl α-d-galactopyranosyl-(1→4)-β-d-galactopyranosyl-(1→3)β-d-galactopyranosyl-(1→3)-β-d-galactopyranosyl-(1→3)-2-acetamido-2-deoxy-α-d-galactopyranoside (**43**)

Compound **43** was prepared from **42** (215 mg, 0.12 mmo) by the same method described for preparation of **18** from **39**. The product was purified by Sephadex LH-20 column chromatography in MeOH to give **43** (107 mg, 90%). ${\left[\alpha \right]}_{D}^{23}$ +96.2 (*c* 1.0, H_2_O). ^1^H-NMR (500 MHz, D_2_O): δ 4.96 (1H, d, *J*_1, 2_ = 3.7 Hz, H-1 of Gal d), 4.88 (1H, d, *J*_1, 2_ = 3.7 Hz, H-1 of GalN), 4.69 (1H, d, *J*_1, 2_ = 7.6 Hz, H-1 of Gal a), 4.66 (1H, d, *J*_1, 2_ = 7.6 Hz, H-1 of Gal c), 4.52 (1H, d, *J*_1, 2_ = 7.9 Hz, H-1 of Gal b), 3.69 (3H, s, OMe), 2.41 (2H, t, -CH_2_-), 2.01 (3H, Ac), 1.66–1.60 (4H, m, 2×-CH_2_-), 1.42–1.32 (2H, m, -CH_2_-). ^13^C-NMR (125 MHz, D_2_O): δ 178.16, 175.04, 104.99 × 2 (C-1 of Gal a, C-1 of Gal b), 104.6 (C-1 of Gal c), 100.9 (C-1 of Gal d), 97.6 (C-1 of GalN), 82.5, 82.3, 78.1, 77.9, 75.7, 75.3, 75.2, 72.7, 71.6, 71.4, 71.1, 70.9, 70.4, 69.7, 69.5, 69.3, 69.2, 69.02, 68.95, 68.4, 61.8, 61.51, 61.45, 61.1, 60.8, 52.7, 49.3, 34.2, 28.7, 25.5, 24.6, 22.6. ESI-HRMS: calcd for C_39_H_67_NO_28_Na, 1020.3747 *m*/*z*; found, 1020.3753 *m*/*z* [M + Na]^+^.

Biotinylated pentasaccharide (**C**)

Compound **C** was prepared from **43** (49 mg, 48.8 μmol) as described for preparation of **B**, yielding 52 mg (85%). ${\left[\alpha \right]}_{D}^{23}$ +85.5 (*c* 1.0, H_2_O). ^1^H-NMR (500 MHz, D_2_O): δ 4.86 (1H, d, *J*_1, 2_ = 4.1 Hz, H-1 of Gal d), 4.88 (1H, d, *J*_1, 2_ = 3.7 Hz, H-1 of GalN), 4.70 (1H, d, *J*_1, 2_ = 7.6 Hz, H-1 of Gal a), 4.66 (1H, d, *J*_1, 2_ = 7.6 Hz, H-1 of Gal c), 4.52 (1H, d, *J*_1, 2_ = 7.8 Hz, H-1 of Gal b), 3.32 (3H, s, OMe). ^13^C-NMR (125 MHz, D_2_O): δ 177.7, 177.6, 175.0, 165.9, 105.0×2 (C-1 of Gal a, C-1 of Gal b), 104.6 (C-1 of Gal c), 100.9 (C-1 of Gal d), 97.6 (C-1 of GalN), 82.6, 82.4, 78.2, 77.9, 75.7, 75.3, 75.2, 72.7, 71.6, 71.4, 71.1, 70.9, 70.4, 69.7, 69.6, 69.2, 69.2, 69.02, 68.96, 68.5, 62.7, 61.8, 61.52, 61.46, 61.1, 60.84, 60.78, 55.9, 49.3, 40.3, 39.2, 39.1, 36.4, 36.1, 28.8, 28.5, 28.3, 25.7, 25.5, 22.6, 20.6. ESI-HRMS: calcd for C_50_H_85_N_5_O_29_SNa, 1274.4949 *m*/*z*; found, 1274.4942 *m*/*z* [M + Na]^+^.

5-(Methoxycarbonyl)pentyl 4,6-di-*O*-acetyl-2,3-di-*O*-benzyl-α-d-galactopyranosyl-(1→4)-2-*O*-benzoyl-3,6-di-*O*-benzyl-β-d-galactopyranosyl-(1→4)-3-*O*-benzoyl-6-*O*-benzyl-2-deoxy-2- (2,2,2-trichloroethoxycarbonyl-amino)-β-d-glucopyranoside (**45**)

To a solution of **44** (100 mg, 64.9 μmol) and methyl 6-hydroxyhexanoate (19 mg, 0.13 mmol) in dry CH_2_Cl_2_ (1 mL) was added powdered MS AW-300 (120 mg), and the mixture was stirred under Ar atmosphere at room temperature for 3 h, then cooled to −20 °C. NIS (63.3 mg, 0.130 mmol) and TfOH (2.73 μL, 13.0 μmol) were added to the mixture, which was stirred at −20 °C for 1 h, then neutralized with Et_3_N. The precipitates were filtered off and washed with CHCl_3_. The combined filtrate and washings were successively washed with saturated aqueous Na_2_S_2_O_3_ and water, dried (MgSO_4_), and concentrated. The product was purified by silica gel column chromatography (2:1 hexane-EtOAc) to give **45** (89 mg, 89%). ${\left[\alpha \right]}_{D}^{24}$ +20.8 (*c* 1.0, CHCl_3_). ^1^H-NMR (500 MHz, CDCl_3_): δ 8.01–7.05 (35H, m, 7×Ph), 5.78 (1H, d, *J* = 9.7 Hz, NH), 5.61 (1H, d, *J* =2.1 Hz, H-4 of Gal b), 5.44–5.93 (2H, m, H-3 of GlcN, H-2 of Gal a), 4.97 (1H, d, *J*_1, 2_ = 3.5 Hz, H-1 of Gal b), 4.63 (1H, d, *J*_1, 2_ = 6.0 Hz, H-1 of Gal a), 4.35 (1H, d, *J*_1, 2_ = 7.0 Hz, H-1 of GlcN), 3.65 (1H, s, OMe), 2.35 (2H, t, -CH_2_-), 1.94 and 1.85 (6H, each s, 2×Ac),1.60–1.50 (4H, m, 2×-CH_2_-), 1.32–1.29 (2H, m, -CH_2_-). ^13^C-NMR (125 MHz, CDCl_3_): δ 174.2, 170.4, 170.2, 165.6, 154.4, 138.5, 138.2, 138.1, 137.9, 137.2, 133.1, 133.0, 130.0, 128.4, 128.3, 128.2, 128.0, 127.6, 127.5, 127.4, 127.3, 101.4, 101.0 (C-1 of Gal a), 100.3 (C-1 of Gal b), 95.4 (C-1 of GlcN), 85.2, 78.5, 76.4, 76.2, 75.1, 74.4, 74.1, 73.9, 73.5, 73.4, 73.0, 72.7, 71.8, 71.6, 71.1, 69.3, 68.0, 67.6, 66.8, 66.5, 62.4, 61.2, 56.3, 51.4, 34.0, 33.9, 33.8, 32.2, 29.6, 29.0, 25.2, 24.5, 24.3, 20.7, 20.6. ESI-HRMS: calcd for C_81_H_88_Cl_3_NO_23_Na, 1570.4710 *m*/*z*; found, 1570.4782 *m*/*z* [M + Na]^+^.

5-(Methoxycarbonyl)pentyl 4,6-di-*O*-acetyl-2,3-di-*O*-benzyl-α-d-galactopyranosyl-(1→4)-2-*O*-benzoyl-3,6-di-*O*-benzyl-β-d-galactopyranosyl-(1→4)-2-acetamido-3-*O*-benzoyl-6-*O*-benzyl-2-deoxy-β-d-glucopyranoside (**46**)

To a solution of **45** (89 mg, 57.6 μmol) in THF—AcOH—Ac_2_O (3:2:1, 4.0 mL) Zn—Cu (0.50 g) was added. The mixture was stirred at room temperature for 30 min. After completion of the reaction, the solid was filtered off. The filtrate was concentrated and purified by silica gel column chromatography (3:1 toluene-acetone) to give **46** (60 mg, 74%). ${\left[\alpha \right]}_{D}^{24}$ +40.1 (*c* 1.0, CHCl_3_). ^1^H-NMR (500 MHz, CDCl_3_) δ 7.99–7.09 (35H, m, 7×Ph), 5.85 (1H, d, *J* = 9.7 Hz, NH), 5.62 (1H, d, *J* =2.1 Hz, H-4 of Gal b), 5.44 (1H, dd, *J*_1, 2_ = 7.9, *J*_2, 3_ = 10.5Hz, H-2 of Gal), 5.35 (1H, t, *J*_1, 2_ = *J*_2, 3_ = 7.9Hz, H-3 of GlcN), 5.00 (d, 1H, *J*_1, 2_ = 3.5 Hz, H-1 of Gal b), 4.62 (d, 1H, H-1 of GlcN), 4.40 (d, 1H, H-1 of Gal a), 3.64 (1H, s, OMe), 2.25 (2H, t, -CH_2_-), 2.03, 1.94 and 1.85 (9H, each s, 3×Ac),1.65–1.48 (4H, m, 2×-CH_2_-), 1.34–1.27 (2H, m, -CH_2_-). ^13^C-NMR (125 MHz, CDCl_3_): δ 174.1, 170.3, 170.2, 170.0, 166.0, 165.2, 138.6, 138.2, 138.0, 137.9, 137.2, 133.2, 130.0, 129.8, 129.7, 129.0, 128.4, 128.2, 128.0, 127.7, 127.6, 127.5, 127.4, 125.3, 101.4 (C-1 of GlcN), 100.9 (C-1 of Gal a), 100.4 (C-1 of Gal b), 78.4, 76.4, 75.4, 75.2, 74.5, 73.6, 73.5, 73.3, 73.1, 72.8, 71.8, 71.7, 71.4, 69.0, 68.4, 67.5, 66.9, 66.6, 61.2, 53.1, 51.4, 33.9, 29.0, 25.4, 24.5, 23.1, 21.4, 20.8, 20.7. ESI-HRMS: calcd for C_80_H_89_NO_22_Na, *m*/*z* 1438.5774; found, *m*/*z* 1438.5842 [M + Na]^+^.

5-(Methoxycarbonyl)pentyl α-d-galactopyranosyl-(1→4)-β-d-galactopyranosyl-(1→4)-2- acetamido-2-deoxy-β-d-glucopyranoside (**47**)

To a solution of **46** (53 mg, 37.7 μmol) in THF (2.0 mL) was hydrogenolysed in the presence of Pd(OH)_2_/C (600 mg) at room temperature for 0.5 h. The mixture was filtered and concentrated, and the residue was acetylated with acetic anhydride (0.3 mL) in pyridine (0.5 mL). After the reaction was quenched with MeOH, toluene was added and co-evaporated several times. The product was purified by silica gel column chromatography (1:1 toluene-acetone) to give an acylated compound (38 mg, 86%). ESI-HRMS: calcd for C_55_H_69_NO_27_Na, *m*/*z* 1198.3955; found, *m*/*z* 1198.4036 [M + Na]^+^. To a solution of this compound in MeOH (1.0 mL), NaOMe (10 mg) was added and the mixture was stirred at 50 °C for 2 h, then neutralized with Amberlite IR 120 [H^+^]. The mixture was filtered off and concentrated. The product was purified by Sephadex LH-20 column chromatography in MeOH to give **47** (19.6 mg, 77%). ${\left[\alpha \right]}_{D}^{24}$ +27.6 (*c* 0.50, MeOH). ^1^H-NMR (500 MHz, CD_3_OD) δ 4.89 (1H, d, *J*_1, 2_ = 3.6 Hz, H-1 of Gal b), 4.37 (1H, d, *J*_1, 2_ = 7.3 Hz, H-1 of Gal a), 4.34 (1H, d, *J*_1, 2_ = 8.1 Hz, H-1 of GlcN), 3.28 (1H, s, OMe), 2.25 (2H, t, -CH_2_-), 1.90 (3H, s, Ac),1.56–1.47 (4H, m, 2×-CH_2_-), 1.33–1.30 (2H, m, -CH_2_-). ^13^C-NMR (125 MHz, CD_3_OD): δ 175.9, 173.5, 105.4 (C-1 of Gal a), 102.6 (C-1 of Gal b, GlcN), 81.4, 79.7, 76.6, 76.5, 74.7, 74.2, 72.8, 72.7, 71.3, 71.0, 70.5, 70.3, 62.7, 62.0, 61.4, 56.9, 52.0, 49.8, 49.5, 49.3, 34.8, 30.2, 26.6, 25.7, 23.0. ESI-HRMS: calcd for C_27_H_47_NO_18_Na, *m*/*z* 696.2691; found, *m*/*z* 696.2714 [M + Na]^+^.

Biotinylated trisaccharide (**D**)

Compound **D** was prepared from **47** (21 mg, 30.9 μmol) as described for preparation of **B**, yielding 13.5 mg (47%). ${\left[\alpha \right]}_{D}^{24}$ +27.6 (*c* 0.50, H_2_O). ^1^H-NMR (500 MHz, D_2_O) δ 4.93 (1H, d, *J*_1, 2_ = 4.5 Hz, H-1 of Gal b), 4.51 (1H, d, *J*_1, 2_ = 8.0 Hz, H-1 of GlcN), 4.50 (1H, d, *J*_1, 2_ = 8.5 Hz, H-1 of Gal a), 2.00 (3H, s, Ac). ^13^C-NMR (125 MHz, D_2_O): δ 177.8, 177.7, 175.0, 166.0, 103.9 (C-1 of Gal a), 101.7 (C-1 of GlcN), 100.9 (C-1 of Gal b), 79.4, 77.9, 76.1, 75.5, 73.2, 72.8, 71.6, 71.5, 70.9, 69.8, 69.6, 69.2, 62.8, 61.1, 61.0, 60.9, 60.7, 56.0, 55.9, 40.4, 39.9, 39.2, 36.5, 36.2, 29.0, 28.6, 28.4, 25.8, 25.6, 25.5, 25.3, 22.9. ESI-HRMS: calcd for C_38_H_65_N_5_O_19_SNa, *m*/*z* 950.3892; found, *m*/*z* 950.3971 [M + Na]^+^.

5-(Methoxycarbonyl)pentyl 2-*O*-benzoyl-3,6-di*-O*-benzyl-4*-O*-chloroacetyl-β-d-galactopyranosyl-(1→3)-2,4,6-tri-*O*-benzoyl-β-d-galactopyranosyl-(1→3)-2-azido-4*-O*-benzyl-2-deoxy-α-d-galactopyranoside (**48**)

To a solution of **10** (468 mg, 0.33mmol)) in dry CH_2_Cl_2_ (6.5 mL) MS AW-300 (500 mg) was added, and the mixture was stirred at room temperature for 2 h, then cooled to −78 °C. Et_3_SiH (160 μL, 0.99 mmol) and PhBCl_2_ (0.15 mL 1.12 mmol) were added, and the mixture was stirred for 45 min, then neutralized with Et_3_N and added to MeOH. The precipitates were filtrated off and washed with CHCl_3_. The combined filtrate and washings were successively washed with water, dried (MgSO_4_), and concentrated. The product was purified by silica gel column chromatography (1:1 hexane—EtOAc) to give **48** (413 mg, 88%). ${\left[\alpha \right]}_{D}^{24}$ +57.1 (*c* 0.50, CHCl_3_). ^1^H-NMR (500 MHz, CDCl_3_): δ 8.07–6.93 (35H, m, 7×Ph), 5.80 (1H, d, *J*_3,4_ = 3.0 Hz, H-4 of Gal b), 5.67 (1H, dd, *J*_1, 2_ = 8.0, *J*_2, 3_ = 10Hz, H-2 of Gal a), 5.12 (1H, d, *J*_1, 2_ = 7.7 Hz, *J*_2, 3_ = 10Hz, H-2 of Gal b), 5.67 (1H, dd, *J*_1, 2_ = 8.0, *J*_2, 3_ = 10Hz, H-2 of Gal a), 4.88 (1H, *J*_1, 2_ = 7.5 Hz, H-1 of Gal a), 4.75 (d, 1H, *J*_1, 2_ = 3.5 Hz, H-1 of GalN), 4.72 (d, 1H, *J*_1, 2_ = 7.5 Hz, H-1 of Gal b), 3.64 (1H, s, OMe), 2.25 (2H, t, -CH_2_-), 1.73–1.46 (4H, m, 2×-CH_2_-), 1.37–1.23 (2H, m, -CH_2_-). ^13^C-NMR (125 MHz, CDCl_3_): δ 174.1, 166.8, 166.0, 164.4, 137.9, 136.8, 133.3, 133.2, 132.7, 132.5, 130.0, 129.6, 129.3, 128.8, 128.6, 128.5, 128.3, 128.2, 128.1, 128.0, 127.9, 127.6, 102.9 (C-1 of Gal a), 101.2 (C-1 of Gal b), 98.1 (C-1 of GalN), 76.2, 75.7, 74.5, 73.7, 72.0, 71.9, 71.3, 70.8, 70.7, 70.5, 70.3, 67.9, 67.5, 67.1, 63.0, 62.3, 59.1, 51.5, 40.6, 33.8, 31.9, 29.7, 29.3, 28.9, 25.6, 24.6, 22.7, 14.1. ESI-HRMS: calcd for C_76_H_78_ClN_3_O_22_Na, *m*/*z* 1442.4663; found, *m*/*z* 1442.4816 [M + Na]^+^.

5-(Methoxycarbonyl)pentyl 2-*O*-benzoyl-3,6-di-*O*-benzyl-4-*O*-chloroacetyl-β-d-galactopyranosyl-(1→3)-2,4,6-tri-*O*-benzoyl-β-d-galactopyranosyl-(1→3)-[4,6-di-*O*-acetyl-2,3-di-*O*-benzyl-α-d-galactopyranosyl-(1→4)-2-*O*-benzoyl-3,6-di-*O*-benzyl-β-d-galactopyranosyl-(1→4)-3-*O*-benzoyl-6-*O*-benzyl-2-deoxy-2-(2,2,2-trichloroethoxycarbonylamino)-β-d-glucopyranosyl-(1→6)]-2-azido-4-*O*-benzyl-2-deoxy-α-d-galactopyranoside (**49**)

Compound **49** was prepared from **44** (298 mg, 0.21mmol) and **48** (355 mg, 0.23mmol) as described for preparation of **7**. The product was purified by silica gel column chromatography (8:7 n-hexane—EtOAc) to give **49** (559 mg, 94%). ${\left[\alpha \right]}_{D}^{24}$ +52.7 (*c* 1.0, CHCl_3_). ^1^H-NMR (500 MHz, CDCl_3_): δ 8.01–6.90 (70H, m, 14×Ph), 5.79–5.07 (7H, m, H-2, 4 of Gal a and Gal b, H-2 of Gal c, H-4 of Gal d), 4.97 (d, 1H, *J*_1, 2_ = 3.5 Hz, H-1 of Gal d), 4.90 (1H, d, *J* = 11.0 Hz, NH), 4.85 (d, 1H, *J*_1, 2_ = 7.5 Hz, H-1 of Gal a), 4.70 (d, 1H, *J*_1, 2_ = 8.0 Hz, H-1 of Gal b), 4.69 (d, 1H, *J*_1, 2_ = 3.5 Hz, H-1 of GalN), 4.62 (d, 1H, *J*_1, 2_ = 7.0 Hz, H-1 of GlcN), 4.58 (d, 1H, *J*_1, 2_ = 8.0 Hz, H-1 of Gal c), 3.59 (1H, s, OMe), 2.23 (2H, t, -CH_2_-), 2.00 and 1.94 (6H, each s, 2×Ac),1.58–1.54 (4H, m, 2×-CH_2_-), 1.31–1.25 (2H, m, -CH_2_-). ^13^C-NMR (125 MHz, CDCl_3_): δ 102.7 (C-1 of Gal a), 101.2 (C-1 of Gal c), 101.0 (C-1 of Gal b), 100.9 (C-1 of GlcN), 100.3 (C-1 of Gal d), 97.8 (C-1 of GalN). ESI-HRMS: calcd for C_150_H_152_Cl_4_N_4_O_42_Na, *m*/*z* 2843.8533; found, *m*/*z* 2843.8428 [M + Na]^+^.

5-(Methoxycarbonyl)pentyl 2-*O*-benzoyl-3,6-di-*O*-benzyl-4-*O*-chloroacetyl-β-d-galacto-pyranosyl-(1→3)-2,4,6-tri-*O*-benzoyl-β-d-galactopyranosyl-(1→3)-[4,6-di-*O*-acetyl-2,3-di-*O*-benzyl-α-d-galactopyranosyl-(1→4)-2-*O*-benzoyl-3,6-di-*O*-benzyl-β-d-galactopyranosyl-(1→4)-2-acetamido-3-*O*-benzoyl-6-*O*-benzyl-2-deoxy-β-d-glucopyranosyl-(1→6)]-2- acetamido-4-*O*-benzyl-2-deoxy-α-d-galactopyranoside (**50**)

Compound **50** was prepared from **49** (200 mg, 70.8 μmol) by the same method described for preparation of **46**. The product was purified by silica gel column chromatography (2:1 toluene-acetone) to give **50** (128 mg, 67%). ${\left[\alpha \right]}_{D}^{24}$ +48.6 (*c* 1.0, CHCl_3_). ^1^H-NMR (500 MHz, CDCl_3_): δ 8.01–6.50 (70H, m, 14×Ph), 5.78–5.06 (7H, m, H-2, 4 of Gal a and Gal b, H-2 of Gal c, H-4 of Gal d), 4.98 (d, 1H, *J*_1, 2_ = 3.0 Hz, H-1 of Gal d), 4.85 (d, 1H, *J*_1, 2_ = 8.0 Hz, H-1 of Gal a), 4.71 (d, 1H, *J*_1, 2_ = 8.5 Hz, H-1 of GlcN), 4.66 (d, 1H, *J*_1, 2_ = 3.5 Hz, H-1 of GalN) 4.60 (d, 1H, *J*_1, 2_ = 7.0 Hz, H-1 of Gal a), 4.23 (d, 1H, *J*_1, 2_ = 7.5 Hz, H-1 of Gal c), 3.61 (1H, s, OMe), 2.23 (2H, t, -CH_2_-), 2.01×2, 1.935 ×2 (12H, each s, 4×Ac),1.66–1.64 (4H, m, 2×-CH_2_-), 1.32–1.25 (2H, m, -CH_2_-). ^13^C-NMR (125 MHz, CDCl_3_): δ 102.1 (C-1 of Gal a), 101.4 (C-1 of Gal c), 101.2 (C-1 of Gal b), 100.9 (C-1 of GlcN), 100.3 (C-1 of Gal d), 96.8 (C-1 of GalN). ESI-HRMS: calcd for C_151_H_157_ClN_2_O_42_Na, *m*/*z* 2727.9797; found, *m*/*z* 2727.9705 [M + Na]^+^.

5-(Methoxycarbonyl)pentyl β-d-galactopyranosyl-(1→3)-β-d-galactopyranosyl-(1→3)-[α-d-galactopyranosyl-(1→4)-β-d-galactopyranosyl-(1→4)-2-acetamido-2-deoxy-β-d-glucopyranosyl-(1→6)]-2-acetamido-2-deoxy-α-d-galactopyranoside (**51**)

To a solution of **50** (90 mg, 33.2 μmol) in MeOH—dioxane (1:1, 3.0 mL) NaOMe (30 mg) was added at room temperature and the mixture was stirred at 40 °C for 3 h, then neutralized with Amberlite IR 120[H^+^]. The mixture was filtered off and concentrated. The residue was purified by silica gel column chromatography using 20:1 CHCl_3_—MeOH as eluent to give the target intermediate (43.7 mg). A solution of the residue in MeOH (6.0 mL) was hydrogenolysed under hydrogen in the presence of 10% Pd/C (50 mg) for 0.5 h at room temperature, then filtered and concentrated. The product was purified by Sephadex LH-20 column chromatography in 1:1 MeOH—H_2_O to give **51** (30.4 mg, 2 steps 76%). ${\left[\alpha \right]}_{D}^{24}$ +39.8 (*c* 1.0, H_2_O). ^1^H-NMR (500 MHz, D_2_O) δ 4.95 (d, 1H, *J*_1, 2_ = 3.8 Hz, H-1 of Gal d), 4.86 (d, 1H, *J*_1, 2_ = 3.8 Hz, H-1 of GalN), 4.61 (d, 1H, *J*_1, 2_ = 7.1 Hz, H-1 of Gal c), 4.55 (d, 1H, *J*_1, 2_ = 8.5 Hz, H-1 of GlcN), 4.53 (d, 1H, *J*_1, 2_ = 7.7 Hz, H-1 of Gal b), 4.52 (d, 1H, *J*_1, 2_ = 7.7 Hz, H-1 of Gal a), 3.70 (1H, s, OMe), 2.42 (2H, t, -CH_2_-), 2.02 and 2.01 (6H, each s, 2×Ac),1.66–1.61 (4H, m, 2×-CH_2_-), 1.43–1.39 (2H, m, -CH_2_-). ^13^C-NMR (125 MHz, CDCl_3_): δ 178.2, 175.1, 174.8, 105.0 (C-1 of Gal a), 104.9 (C-1 of Gal c), 103.9 (C-1 of Gal b), 102.1 (C-1 of GlcN), 100.9 (C-1 of Gal d), 97.4 (C-1 of GalN), 82.6, 79.5, 78.0, 77.9, 76.1, 75.7, 75.5, 75.3, 73.3, 73.2, 72.8, 71.7, 71.6, 71.5, 70.7, 70.5, 69.9, 69.8, 69.6, 69.2, 69.0, 68.0, 61.6, 61.2, 61.0, 60.7, 55.8, 52.8, 49.3, 34.3, 30.9, 28.7, 25.7, 24.7, 22.9, 22.6. ESI-HRMS: calcd for C_47_H_80_N_2_O_33_Na, *m*/*z* 1223.4541; found, *m*/*z* 1223.4639 [M + Na]^+^.

Biotinylated hexasaccharide (**E**)

Compound **E** was prepared from **51** (16 mg, 13.2 μmol) as described for preparation of **B**, yielding 15.4 mg (79%). ${\left[\alpha \right]}_{D}^{24}$ +27.9 (*c* 0.50, H_2_O). ^1^H-NMR (500 MHz, D_2_O): δ 4.95 (d, 1H, *J*_1, 2_ = 3.5 Hz, H-1 of Gal d), 4.86 (d, 1H, *J*_1, 2_ = 3.5 Hz, H-1 of GalN), 4.60 (d, 1H, *J*_1, 2_ = 8.0 Hz, H-1 of Gal c), 4.55 (d, 1H, *J*_1, 2_ = 8.0 Hz, H-1 of GlcN), 4.53 (d, 1H, *J*_1, 2_ = 8.0 Hz, H-1 of Gal b), 4.52 (d, 1H, *J*_1, 2_ = 8.0 Hz, H-1 of Gal a). ^13^C-NMR (125 MHz, D_2_O): δ 180.1, 177.9, 177.7, 175.1, 174.8, 166.0, 105.1 (C-1 of Gal a), 105.0 (C-1 of Gal c), 103.9 (C-1 of Gal b), 102.1 (C-1 of GlcN), 100.9 (C-1 of Gal d), 97.5 (C-1 of GalN), 82.7, 79.5, 77.9, 76.1, 75.7, 75.5, 75.3, 73.3, 73.2, 72.8, 71.7, 71.6, 71.5, 70.5, 70.0, 69.8, 69.6, 69.2, 68.1, 62.7, 61.6, 61.2, 61.0, 90.9, 56.0, 55.8, 49.3, 40.4, 40.0, 39.3, 36.5, 30.8, 30.5, 28.9, 28.6, 28.4, 25.9, 25.8, 25.5, 23.0, 22.7 ESI-HRMS: calcd for C_58_H_98_N_6_O_34_SNa, *m*/*z* 1477.5742; found, *m*/*z* 1477.5882 [M + Na]^+^.

### 4.2. Serum Samples

Serum samples of 60 and 45 patients confirmed to have AE and CE, respectively, and those of 60 healthy individuals, which are kept in Hokkaido Institute of Public Health, were used for ELISA assay under the approval of the institute.

### 4.3. ELISA Protocol

ELISA was performed using as previously described [12] with some modifications. The oligosaccharides in H_2_O (13 pmol per well) were placed in the wells of flat-bottomed microplates (Streptavidin C96, No. 236001; Nunc, Roskilde, Denmark) coated with streptavidin and incubated for 1 h at 37 °C. After removal of the solution, the wells were washed with 0.05% Tween-PBS (250 μL per well). Serum samples diluted 1:250 with 0.05% Tween-PBS (200 μL per well) were then added to the wells and incubated overnight at 4 °C. After removal of the serum and washing with 0.05% Tween-PBS, 200 μL of anti-human IgG/HRP (P0214; DakoCytomation, Denmark; 1:1000 in 0.05% Tween-PBS) was added, and the microplate was incubated for 1 h at 37 °C. After the washing of the wells, bound antibodies were detected by the addition of ABTS peroxidase substrate solution (KPL, Gaithersburg, MD, USA, 200 μL per well). After the incubation period of 8 min at 37 °C, the reaction was stopped by the addition of 1% SDS, and the absorbance (A) values were read at 405 nm on a microplate reader (Model 680; BIORAD, Hercules, CA, USA).

## Data Availability

Not applicable.

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
