# Peer review of "Synthesis of the Carbohydrate Moiety of Glycoproteins from the Parasite Echinococcus granulosus and Their Antigenicity against Human Sera"

_molecules, 2021, doi:10.3390/molecules26185652_

Round 1

Reviewer 1 Report

In this manuscript, the authors have synthesized 5 biotin-labled oligosaccharide portions containing the carbohydrate moiety of glycoprotein from Echinococcus granulosus (A, B, C , D and E), and examined the antigenicity of these five oligosaccharides by an enzyme linked immunosorbent assay (ELISA). They have found that compounds B, D and E showed good serodiagnostic potential for alveolar echinococcosis (AE). I recommend it to be published in this journal after the supporting information is attached. I am surprised that I did not see any supporting information including all the NMR spectrum. Some other comments and suggestions:

  1. For the synthesis of compond 2, the authors can try the organotin-catalyzed benzylation method (Adv. Synth. Catal. 2014, 356, 1735), which is perhaps more convenient and efficient.
  2. For Scheme 2, since the authors have satated that the benzylidene acetal was not removed under 10% Pd/C in acetic acid, I don't understand why conditions 2) and 3) were still included (from 11 to 14).
  3. For Scheme 7 and 8, why didn't the authors use the trichloroacetimidate donor of 34 directly, but use 36 as the donor? 
  4. In experiment parts, I wondered why the HRMS found values of compounds 24 - 35, 41, 42, and 46-51 have large deviations (0.01-0.02) from the calcd values.

Reviewer 2 Report

This work is focused on the synthesis of the carbohydrate moiety of glycoproteins from the parasite Echinococcus granulosus and the study of their antigenicity against human sera.

Authors have experience in this line. They have previously described and employed the methodology for the preparation of a set of oligosaccharide-biotin conjugates (references 8, 11) and studied their antigenicity at E. multilocularis (references 7, 12 and 13).

In this work authors continue their research with the logical next step: the extension to other novel conjugates (in this paper they have prepared hexasaccharide-biotin conjugate among others). The oligosaccharide sequence of these conjugates is based on recent studies of related cestode Echinococcus granulosus (reference 21), that causes cystic echinococcosis (CE) in intermediate hosts like humans.

Related to the Chemistry part, the synthetic strategy developed for each target conjugate are well described and the complete information about the steps and all the problems that authors had to manage during the synthesis are detailed, the schemes and figures are clear and all new compounds are fully characterized.

Concerning their evaluation, the oligosaccharide-biotin conjugates were prepared to study the antigenicity of the compounds to detect antibodies in patient sera infected with E. granulosus the cause of CE. They have found no response in CE, but three glycoconjugates showed good serodiagnostic potential to recognize antibodies in AE patients.

This manuscript is, in my opinion, an example of a well conducted research in carbohydrate chemistry field. The whole manuscript is well written, redacted and its reading flows very well.

In general terms I  recommend it for publication in Molecules with a few suggestions to be attended before its publication (minor revisions)

Line 51: the text should be: in intermediate hosts like humans (in plural)

Figure 1: I do not clearly understand the meaning of the arrow that connects Compounds E and D in the description of the synthetic approach.

Line 66: The Figure caption should be: Synthetic target oligosaccharides from E. granulosus (in plural)

Line 72: The Figure caption should be: Synthetic strategy of tri-, tetra and pentasaccharides (in plural)

Line 107: The Scheme 1 caption should be:. Preparation of tetrasaccharide derivative 13

Line 152: The Scheme 4. caption should be: Preparation of trisacccharide derivatives (in plural)

Line 180: The Scheme 6.caption should be:  Preparation of pentasaccharide derivative 31 (The number of the compound in bold)

Line 202: The Scheme 7 caption should be:. Preparation of disaccharide donor 36 (the number of the compound in bold, in the text it appears in green)

Line 312: The name of compound 5: The number five in the name should not appear in bold.

In summary, this is an interesting piece of work, and I encourage authors to continue this research

Round 2

Reviewer 1 Report

Accept in present form